# MTL-UE: Learning to Learn Nothing for Multi-Task Learning

**Yi Yu** [1 2]  **Song Xia** [2]  **Siyuan Yang** [3]  **Chenqi Kong** [2]  **Wenhan Yang** [† 4]  **Shijian Lu** [3]  **Yap-Peng Tan** [2]  **Alex C. Kot** [2]

## Abstract

Most existing unlearnable strategies focus on preventing unauthorized users from training single-task learning (STL) models with personal data. Nevertheless, the paradigm has recently shifted towards multi-task data and multi-task learning (MTL), targeting generalist and foundation models that can handle multiple tasks simultaneously. Despite their growing importance, MTL data and models have been largely neglected while pursuing unlearnable strategies. This paper presents MTL-UE, the first unified framework for generating unlearnable examples for multi-task data and MTL models. Instead of optimizing perturbations for each sample, we design a generator-based structure that introduces label priors and class-wise feature embeddings which leads to much better attacking performance. In addition, MTL-UE incorporates intra-task and inter-task embedding regularization to increase inter-class separation and suppress intra-class variance which enhances the attack robustness greatly. Furthermore, MTL-UE is versatile with good supports for dense prediction tasks in MTL. It is also plug-and-play allowing integrating existing surrogate-dependent unlearnable methods with little adaptation. Extensive experiments show that MTL-UE achieves superior attacking performance consistently across 4 MTL datasets, 3 base UE methods, 5 model backbones, and 5 MTL task-weighting strategies. Code is available at https://github.com/yuyi-sd/MTL-UE.

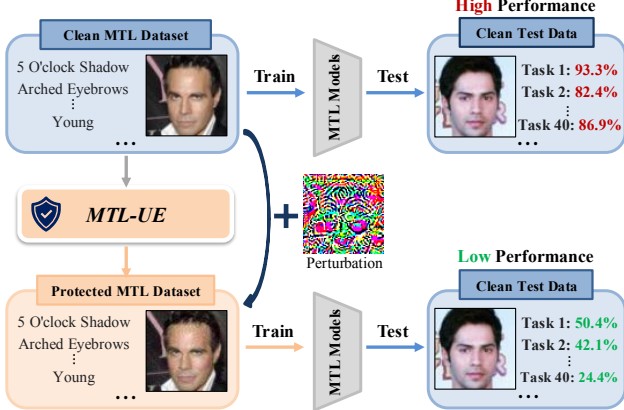

*Figure 1.* Illustration of MTL-UE to prevent unauthorized training of MTL models on datasets like CelebA (Liu et al., 2015), having 40 binary attribute classifications. MTL-UE adds invisible, sample-specific perturbations to transform a clean dataset into a protected one, leading to poor test performance of trained MTL models.

## 1. Introduction

Multi-task learning (MTL) (Caruana, 1993; Guo et al., 2020) is a branch of machine learning that tackles multiple tasks simultaneously, making it a more practical approach than single-task learning (STL). For instance, autonomous vehicles (Achituve et al., 2024) need to detect objects, track vehicles, monitor lanes, and estimate free space in real time. MTL trains a single model to handle multiple tasks, reducing the need for separate models. By leveraging shared data across tasks, MTL lowers computational costs and improves generalization (Baxter, 2000), making it essential in fields like vision (Liu et al., 2019; Misra et al., 2016), NLP (Chen et al., 2021; Fontana et al., 2024), autonomous driving (Chowdhuri et al., 2019; Chen et al., 2018), and recommendation systems (Hadash et al., 2018).

Deep neural networks have achieved impressive success across machine learning tasks (Hu et al., 2024; Yang et al., 2024; Jin et al., 2025a;b; Zhang et al., 2025), but also raised growing AI security concerns (Yu et al., 2022b; 2023; 2024b; 2025; Liu et al., 2025a;b). As large-scale models become more prevalent, massive data is likely to be scraped from the web and incorporated into training datasets, naturally raising concerns about the unauthorized use of personal informa-

---

[†]Corresponding author  [1]Rapid-Rich Object Search Lab, Interdisciplinary Graduate Programme, Nanyang Technological University, Singapore [2]School of Electrical and Electronic Engineering, Nanyang Technological University, Singapore [3]College of Computing and Data Science, Nanyang Technological University, Singapore [4]PengCheng Laboratory, Shenzhen, China. Correspondence to: Yi Yu <yuyi0010@e.ntu.edu.sg>, Wenhan Yang <yangwh@pcl.ac.cn>.

*Proceedings of the $42^{nd}$ International Conference on Machine Learning*, Vancouver, Canada. PMLR 267, 2025. Copyright 2025 by the author(s).

tion for training DNNs (Burt, 2020; Vincent, 2019). This has led to efforts to develop defenses that prevent DNNs from exploiting private data. In STL, privacy protection methods (Feng et al., 2019; Sun et al., 2024) have been widely explored. These methods apply carefully crafted perturbations to images to compromise the generalization of models, commonly referred to as unlearnable examples (UE) (Huang et al., 2021; Yu et al., 2024a;c), and are also known as perturbative availability (Liu et al., 2023) or indiscriminate poisoning attacks (He et al., 2023). Models trained on these UE often capture spurious features, which are patterns added to the data that are irrelevant to the actual task. While UE have been extensively studied for STL, their use in complex MTL scenarios as well as the related datasets, *e.g.,* NYUv2 (Nathan Silberman & Fergus, 2012) with semantic segmentation, depth estimation, and normal estimation tasks, remains a challenge. These include managing the heightened complexity of perturbations designed to introduce spurious features across a greater number of tasks simultaneously, and dealing with more complex tasks that extend beyond classification.

In this work, we propose **MTL-UE**, a framework for generating effective UE for MTL, with the goal of degrading the performance of all tasks in both MTL and STL models, as shown in Fig. 1. We first conduct a straightforward empirical benchmark analysis of existing methods to reveal our motivations. We implement several baseline UE methods, including surrogate-free methods using predefined shortcut patterns as class-wise perturbations, and surrogate-dependent methods relying on surrogate models to optimize sample-wise perturbations. Our initial findings suggest that surrogate-dependent methods underperform for both MTL and STL models, due to poor control over intra-class variance caused by independent optimization for each sample. Patch-based AR (Sandoval-Segura et al., 2022), as shown in Sec. 4.2, using class-wise perturbations in task-specific patches, performs better with lower intra-class variance but loses effectiveness as tasks increase, likely due to smaller patch sizes and limited representation capacity.

Building on these insights, we propose a plug-and-play framework to address these challenges by generating UE through class-wise feature embedding injections. By incorporating task label priors via embeddings, we narrow the perturbation searching space from $\|\boldsymbol{\delta}\|_\infty \leq \frac{8}{255}$ to the decoder's output space, resulting in lower intra-class variance. This approach effectively combines spurious features from multiple tasks into a unified perturbation. In addition to the generator's structural design, we introduce intra-task and inter-task embedding regularization (Intra-ER & Inter-ER) to improve inter-class distance and minimize feature space redundancy, further improving the attack's effectiveness. In summary, our contributions are outlined below:

• To the best of our knowledge, we propose **MTL-UE**, the first plug-and-play framework for generating UE on MTL datasets, effective against both MTL and STL models, and compatible with any surrogate-dependent UE methods.

• MTL-UE comprises an encoder-decoder network paired with learnable class-wise feature embeddings, which lower the intra-class variance of spurious features for each task. The addition of intra-task and inter-task embedding regularization further enhances performance.

• MTL-UE can extend beyond MTL classifications to multiple dense prediction tasks by using task-specific embedding modules to map task labels to embeddings.

• Experiments on 4 MTL datasets, 3 base UE methods, 5 backbones, and 5 MTL task-weighting strategies show consistent improvements of MTL-UE in attacking performance. Moreover, MTL-UE supports partial protection, making some tasks unlearnable while keeping others learnable.

## 2. Related Work

**Data Poisoning.** Data poisoning attacks (Barreno et al., 2010; Goldblum et al., 2022) manipulate training data to disrupt the test-time performance of models, and are categorized into integrity attacks and availability attacks. Integrity attacks, such as backdoor attacks (Gu et al., 2017; Schwarzschild et al., 2021), trigger malicious behavior with specific inputs, while availability attacks degrade model performance on test sets (Biggio et al., 2012; Xiao et al., 2015). Typically, they inject poisoned data into the clean training set, where a small portion of the samples, with unrestricted changes, are added (Koh & Liang, 2017; Zhao & Lao, 2022; Lu et al., 2023). Though malicious, these samples are often detectable and have a limited overall impact.

**Unlearnable Examples (UE).** UE (Huang et al., 2021; Zhang et al., 2023; Zhu et al., 2024a;b; Liu et al., 2024a;b; Chen et al., 2024; Qin et al., 2023; 2024; Meng et al., 2024; Lin et al., 2024; Wang et al., 2024; 2025) is an emerging approach, where subtle modifications, such as bounded perturbations $\|\boldsymbol{\delta}\|_\infty \leq \frac{8}{255}$, are applied across the entire training dataset without altering the correct labels. This method shows potential for data protection, resulting in models performing close to random guessing on clean test data. EM (Huang et al., 2021) applies error-minimizing noise, while NTGA (Yuan & Wu, 2021) generates noise via neural tangent kernels. TAP (Fowl et al., 2021) uses targeted adversarial examples as UE, and REM (Fu et al., 2022) targets adversarial training (AT) (Madry et al., 2018). LSP (Yu et al., 2022a) and AR (Sandoval-Segura et al., 2022) are surrogate-free UE. OPS (Wu et al., 2023) uses one-pixel shortcuts to improve robustness against AT and strong augmentations.

**Multi-task Learning (MTL).** MTL focuses on learning multiple tasks in a joint manner, typically using a shared encoder with task-specific heads for each task (Ruder, 2017; Zhang & Yang, 2021; Sener & Koltun, 2018; Standley et al., 2020; Fifty et al., 2021; Navon et al., 2022). A major focus in MTL research is the optimization process. The usual method uses linear scalarization (LS) along with a grid or random search to find the best weight vectors (Lin et al., 2019). To address task balancing and conflict resolution, strategies fall into loss-based and gradient-based (Dai et al., 2023). Loss-based methods assign task weights based on factors like task difficulty (Guo et al., 2018), random weights (Lin et al., 2022), geometric mean of losses (Yun & Cho, 2023), or uncertainty (Kendall et al., 2018). Gradient-based approaches adjust gradients directly, *e.g.,* PCGrad (Yu et al., 2020) projects gradients to prevent conflicts, Aligned-MTL (Senushkin et al., 2023) aligns gradient components, and FairGrad (Ban & Ji, 2024) adopts utility maximization.

## 3. Preliminaries

UE (Huang et al., 2021; Fowl et al., 2021; Wang et al., 2024) leverage clean-label data poisoning to trick DNNs into learning minimal useful knowledge from the data, thereby achieving the objective of data protection. Let $\mathcal{T}$ and $\mathcal{D}$ denote the clean training and test datasets, respectively. A model $F(\cdot; \theta)$ trained on $\mathcal{T}$ typically performs well on $\mathcal{D}$. UE aims to transform $\mathcal{T}$ into an unlearnable dataset $\mathcal{P}$, causing $F(\cdot; \theta)$ trained on $\mathcal{P}$ to perform poorly on $\mathcal{D}$.

In a $C$-class image classification, $\mathcal{T} = \{(\boldsymbol{x}_i, y_i)\}_{i=1}^{N}$ contains $N$ samples, where $\boldsymbol{x}_i \in \mathcal{R}^d$ are inputs, and $y_i \in \{1, \ldots, C\}$ are labels. $\mathcal{P}$ is crafted by adding perturbations $\boldsymbol{\delta}_i$ to each $\boldsymbol{x}_i$, such that $\mathcal{P} = \{(\boldsymbol{x}_i + \boldsymbol{\delta}_i, y_i)\}_{i=1}^{N}$. Perturbations $\boldsymbol{\delta} \in \mathcal{S}$ are constrained to maintain visual imperceptibility, where $\mathcal{S}$ denotes the feasible region, *e.g.,* $\|\boldsymbol{\delta}\|_{\infty} \leq \frac{8}{255}$. The attacker's success is measured by the accuracy of $F$ trained on $\mathcal{P}$ when evaluated on $\mathcal{D}$. UE methods can be classified into two categories based on whether a surrogate model is required for optimizing the perturbations.

**Surrogate-free methods** (Yu et al., 2022a) do not rely on a surrogate model, and instead utilize predefined shortcut patterns as class-wise perturbations. $\boldsymbol{\delta}_i$ added to each sample $\boldsymbol{x}_i$ depend solely on its label $y_i$, *i.e.,* $\boldsymbol{\delta}_i = \boldsymbol{\delta}(y_i)$.

**Surrogate-dependent methods** optimize sample-wise perturbations $\boldsymbol{\delta}_i$ for each data $(\boldsymbol{x}_i, y_i)$ using surrogate models. Most follow variations of error-minimizing (EM) (Huang et al., 2021) or error-maximizing (AP) (Fowl et al., 2021). EM constructs $\boldsymbol{\delta}_i$ by solving the bi-level optimizations:

$$\min_{\theta} \sum_{(\boldsymbol{x}_i, y_i) \in \mathcal{T}} \Big[ \min_{\boldsymbol{\delta}_i} \mathcal{L}(F'(\boldsymbol{x}_i + \boldsymbol{\delta}_i; \theta), y_i) \Big], \text{ s.t. } \|\boldsymbol{\delta}_i\|_p \leq \epsilon, \quad (1)$$

where $F'$ is the surrogate model. Typically, the inner minimization employs the first-order optimization approach

PGD (Madry et al., 2018), and the outer one optimizes the parameters using optimizers such as SGD. In contrast, AP constructs $\boldsymbol{\delta}_i$ to maximize the loss of the pretrained $F'$ on clean dataset $\mathcal{T}$, *i.e.* generating adversarial examples:

$$\max_{\boldsymbol{\delta}_i} \sum_{(\boldsymbol{x}_i, y_i) \in \mathcal{T}} \big[ \mathcal{L}(F'(\boldsymbol{x}_i + \boldsymbol{\delta}_i; \theta^*), y_i) \big], \quad \text{s.t. } \|\boldsymbol{\delta}_i\|_p \leq \epsilon. \quad (2)$$

**Multi-task learning (MTL)** enhances the performance of several related tasks by training them simultaneously. The training samples are typically tuples consisting of a shared input for all tasks and the labels for $K$ tasks, *i.e.,* $\mathcal{T}_{\text{MTL}} = \{(\boldsymbol{x}_i, \{y_i^k\}_{k=1}^{K})\}_{i=1}^{N}$, where $N$ is the number of training samples. Our focus is on the scenario where the input are consistent across tasks. When the input varies for each task, it is often referred to as multi-domain learning (Royer et al., 2023). In such cases, UE for each domain dataset can usually be constructed independently. Common architectures for MTL (Achituve et al., 2024) have a shared encoder $f(\cdot; \theta_f)$ and task-specific linear heads $g^k(\cdot; \theta_{g^k})$. A MTL problem involves a set of $K$ tasks with a loss vector:

$$\min_{\{\theta_f, \{\theta_{g^k}\}_{k=1}^{K}\}} \mathcal{L} = (\mathcal{L}_1(\theta_f, \theta_{g^1}), \cdots, \mathcal{L}_K(\theta_f, \theta_{g^K}))^{\top}, \quad (3)$$

where $\mathcal{L}_k(\theta_f, \theta_{g^k})$ is the loss of the $k$-th task. An MTL algorithm seeks to optimize all tasks simultaneously by leveraging the shared structure and information across them.

## 4. Methodology

### 4.1. Problem Formulation

Previous UE (Feng et al., 2019; Huang et al., 2021) mainly target STL with $K = 1$ task. However, practical scenarios often involve multi-task datasets. This work aims to develop UE for these datasets, ensuring better data protection against both MTL and STL models, with the following goals[1]:

• **Effectiveness against MTL models**: For a MTL model $F_{\text{MTL}} = \{f, \{g^k\}_{k=1}^{K}\}$ trained on the poisoned multi-task dataset $\mathcal{P}_{\text{MTL}} = \{(\boldsymbol{x}_i + \boldsymbol{\delta}_i, \{y_i^k\}_{k=1}^{K})\}_{i=1}^{N}$, the objective is to maximize the loss on clean test data for all tasks, *i.e.,* maximizing $\sum_{k=1}^{K} \mathcal{L}_k(\boldsymbol{x}, y^k; \theta_f, \theta_{g^k})$.

• **Effectiveness against STL models**: For any $k$-th task and a STL model $F_{\text{STL}}^k = \{f, g^k\}$ trained on the poisoned dataset $\mathcal{P}_{\text{MTL}}^k = \{(\boldsymbol{x}_i + \boldsymbol{\delta}_i, y_i^k)\}_{i=1}^{N}$ (the $k$-th attribute of $\mathcal{P}_{\text{MTL}}$), the objective is to maximize the loss $\mathcal{L}_k(\boldsymbol{x}, y^k; \theta_f, \theta_{g^k})$.

### 4.2. Baseline methods for UE against MTL

This section presents several MTL-specific baseline UE methods, starting with surrogate-dependent methods that typically use the MTL model itself as the surrogate model.

---

[1]We will omit "MTL" in $\mathcal{T}_{\text{MTL}}$ and $\mathcal{P}_{\text{MTL}}$ in later sections.

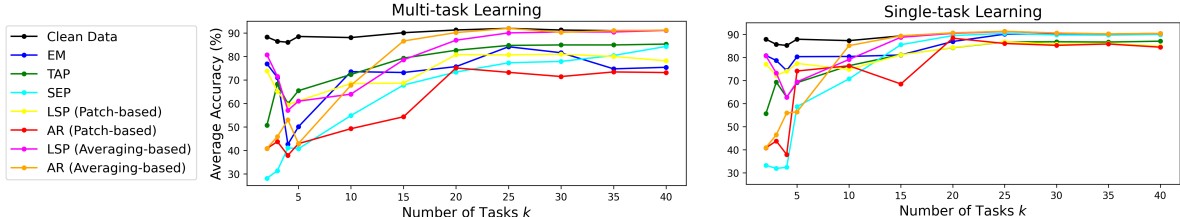

*Figure 2.* Performance of UE (Accuracy ↓) Vs. the number of tasks on the CelebA (Liu et al., 2015) for both MTL and STL models.

*Table 1.* Intra-class std of the features for various UE methods. (P) and (A) are the patch and averaging based. All 40 tasks are used.

| UEs → | Clean | EM | TAP | SEP | LSP (P) | AR (P) | LSP (A) | AR (A) |
|---|---|---|---|---|---|---|---|---|
| Average | 5.357 | 2.191 | 3.656 | 6.719 | 1.828 | 1.733 | 5.264 | 5.180 |
| Maximum | 91.97 | 82.13 | 103.12 | 96.24 | 34.84 | 20.59 | 93.50 | 103.41 |

Next, we consider surrogate-free methods and class-wise perturbations. Treating each combination $\{y^k\}_{k=1}^K$ as a distinct class, where $y^k \in \{1, \ldots, C_k\}$, yields a total of $\prod_{k=1}^K C_k$ class-wise perturbations, which grows exponentially with $K$. This reduces the average number of data points per perturbation to $\frac{N}{\prod_{k=1}^K C_k}$, complicating the creation of spurious relationships between perturbations and target combinations. To address this, we propose generating separate sets of perturbations for each task, $\{\{\boldsymbol{\delta}_{y^k}^k\}_{y^k=1}^{C_k}\}_{k=1}^K$, and then combining them into a final perturbation. We explore two strategies for this combination: (1) **Averaging-based**: using the mean of $\{\boldsymbol{\delta}_{y^k}^k\}_{k=1}^K$ as $\boldsymbol{\delta}$; and (2) **Patch-based**: adding $\{\boldsymbol{\delta}_{y^k}^k\}_{k=1}^K$ to task-specific non-overlapping patches of $\boldsymbol{x}$.

We assess UE effectiveness across varying task numbers on CelebA (Liu et al., 2015), a facial dataset with 40 binary classifications. We evaluate 5 UE methods: EM (Huang et al., 2021), TAP (Fowl et al., 2021), and SEP (Chen et al., 2023) are surrogate-dependent using surrogate MTL models with uniform task-weighting, while LSP (Yu et al., 2022a) and AR (Sandoval-Segura et al., 2022) are surrogate-free.

**Effectiveness of UE Vs. the number of tasks.** To examine the impact of task quantity on the effectiveness of baseline UE, we select the first $k$ tasks, generate the corresponding UE, and train MTL and STL models. The experimental results as shown in Fig. 2 highlight several key findings:

**1.** STL models are more robust to UE than MTL models.

**2.** As $k$ increases, the performance of most UE initially increases on MTL and STL models, then declines, with near total failure on STL when all tasks are included.

**3.** Patch-based AR excels in both MTL and STL models.

The first observation arises from MTL models sharing representations across tasks, allowing them to more effectively capture similar shortcut patterns in UE and enhancing their focus on spurious features over benign ones. The following explanations address the remaining two points.

As discussed in (Yu et al., 2024a), spurious features with lower intra-class variance and greater inter-class distance are more effective for attacks. We denote the encoder's features in MTL models as $\boldsymbol{z} = [z_1, z_2, \ldots, z_D] \in \mathbf{R}^D$. For each $z_d$, we compute the average relative intra-class std (standard deviation) across all classes and tasks as $\frac{1}{\sum_k C_k} \sum_k \sum_{y^k} \left[ \frac{\mathbf{Std}[z_d|y^k]}{\mathbf{E}[z_d|y^k]} \right]$. The average and maximum values across the $D$ dimensions are in Tab. 1. Note that features are from the MTL models using the poisoned dataset.

Our results show that patch-based AR has the lowest intra-class variance, with another patch-based approach that ranks second. This may be due to class-wise perturbations being applied to distinct, task-specific patches, thereby minimizing intra-class variance across locations. These lower intra-class variances likely contribute to the effectiveness. However, as $k$ increases, the reduction in patch size constrains the perturbations' ability as effective shortcuts, ultimately leading to a performance decline. In contrast, averaging-based AR and LSP have higher intra-class variance and perform worse than patch-based methods, as aggregating perturbations across tasks can introduce conflicts, leading to suboptimal results.

Surrogate-dependent methods like EM, TAP, and SEP optimize perturbations individually for each sample, which limits control over intra-class variance. This limitation can reduce performance, particularly as $k$ increases, since perturbations must serve as shortcuts across multiple tasks, complicating the optimization. As shown in Tab. 1 and Fig. 2, most UE methods have higher intra-class variance and lower performance compared to patch-based AR and LSP, although EM achieves the second-best attack performance among MTL models and the lowest intra-class variance among surrogate-dependent methods. Overall, we see that LSP (A), AR (A), TAP, and SEP achieve intra-class variance comparable to that of clean data, resulting in limited effectiveness in attacking both MTL and STL models.

### 4.3. MTL-UE: a plug-and-play UE method for MTL

**What is missing in existing UE methods?** Based on our observations in Sec. 4.2, we identify: (1) Surrogate-dependent UE perform poorly due to uncontrolled intra-class variance caused by **individual sample optimization**, and the **pixel-level search space** which is hard to control; (2) Surrogate-

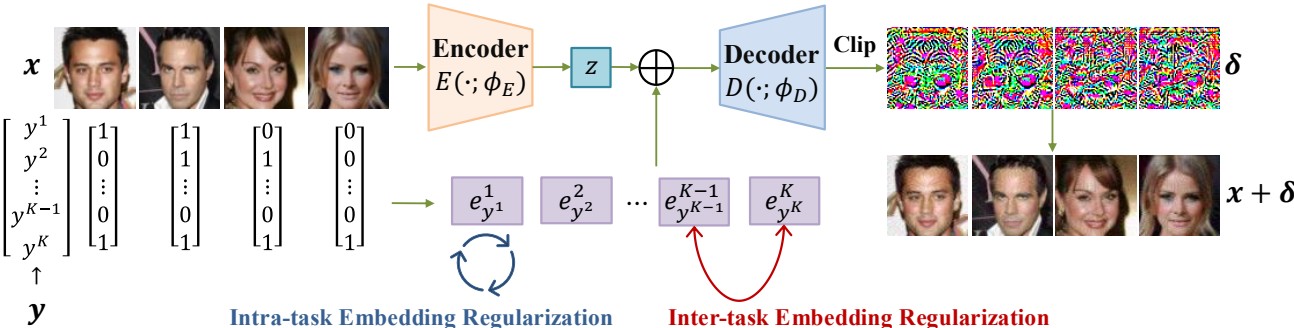

*Figure 3.* Visual depiction of MTL-UE, concatenating the task-specific class-wise embeddings with latents to generate UE for MTL data.

---

**Algorithm 1** Optimization of the UE Generator in MTL-UE

**Input:** Surrogate model $F'_{\text{MTL}} = \{f, \{g^k\}_{k=1}^K\}$, encoder $E(\cdot; \phi_E)$, decoder $D(\cdot; \phi_D)$, embeddings $\{\{e_i^k\}_{i=1}^{C_k}\}_{k=1}^K$, clean multi-task dataset $\mathcal{T} = \{(\boldsymbol{x}_i, \{y_i^k\}_{k=1}^K)\}_{i=1}^N$, epochs $R$, Adam optimizer, weights $\lambda_1$ & $\lambda_2$, Train-surrogate, iterations $I$, bound $\epsilon$, loss function of the base UE method $\mathcal{L}_b$

**Output:** Optimized $E(\cdot; \phi_E)$, $D(\cdot; \phi_D)$, $\{\{e_i^k\}_{i=1}^{C_k}\}_{k=1}^K$
# Initialize surrogate model if no training is needed in the Alg.
**if** not Train-surrogate **then**
  Initialize $F'_{\text{MTL}}$ with pretrained weights on $\mathcal{T}$
**end if**
**for** epoch $\leftarrow 1$ **to** $R$ **do**
  # Optimize the perturbation generator
  **for** image batch $(\boldsymbol{x}, \boldsymbol{y} = \{y^k\}_{k=1}^K) \in \mathcal{T}$ **do**
    $\boldsymbol{z} = E(\boldsymbol{x}; \phi_E)$, $\boldsymbol{\delta} = \text{Clip}(D([\boldsymbol{z}, e_{y^1}^1, \ldots, e_{y^K}^K]; \phi_D), -\epsilon, \epsilon)$
    $\mathcal{L} = \mathcal{L}_b(F'_{\text{MTL}}, \boldsymbol{x} + \boldsymbol{\delta}, \boldsymbol{y}) + \lambda_1 \cdot \mathcal{L}_{Intra} + \lambda_2 \cdot \mathcal{L}_{Inter}$
    Minimize $\mathcal{L}$ with Adam to update $\phi_E$, $\phi_D$, and $e_i^k$
  **end for**
  **if** Train-surrogate **then**
    # Optimize $F'_{\text{MTL}}$ if training is needed
    **for** epoch $\leftarrow 1$ **to** $I$ **do**
      Sample image batch $(\boldsymbol{x}, \boldsymbol{y} = \{y^k\}_{k=1}^K) \in \mathcal{T}$
      $\boldsymbol{\delta} = \text{Clip}(D([E(\boldsymbol{x}; \phi_E), e_{y^1}^1, \ldots, e_{y^K}^K]; \phi_D), -\epsilon, \epsilon)$
      Train $F'_{\text{MTL}}$ with Adam on $(\boldsymbol{x} + \boldsymbol{\delta}, \boldsymbol{y})$
    **end for**
  **end if**
**end for**

---

free UEs face challenges primarily due to the **imperfect fusion** of class-wise perturbations from different tasks.

This raises the question: *Can we reduce the pixel-level searching space to learn task-specific class-wise spurious features and use an integration network to merge them into a unified perturbation?*

To address these, we propose **MTL-UE**, which generates UEs by injecting class-wise features to manage intra-class variance and combining spurious features from multiple tasks into a unified perturbation. The framework is shown in Fig. 3. For any input $\boldsymbol{x}$, the encoder $E(\cdot; \phi_E)$ maps it to a la-

tent representation $\boldsymbol{z}$. Based on its labels $\{y^k\}_{k=1}^K$, the corresponding learnable class-wise feature embeddings $\{e_{y^k}^k\}_{k=1}^K$ are selected. The embeddings and $\boldsymbol{z}$ are concatenated, and fed into a decoder $D(\cdot; \phi_D)$ to generate the final perturbations for $\boldsymbol{x}$. To regulate the amplitude of the perturbations, a clip operation is applied following the decoder. By shifting from direct perturbation searching to spurious features and integration network learning, **MTL-UE can seamlessly integrate with any surrogate-dependent method**.

Besides MTL-UE's structural design, we introduce embedding regularizations to further improve attack performance:

● **Intra-task ER (Intra-ER).** This term minimizes the cosine similarity between embeddings within each task, ensuring diversity among the embeddings:

$$\mathcal{L}_{Intra} = \frac{2}{\sum_{k=1}^K C_k(C_k - 1)} \sum_{k=1}^K \sum_{m=1}^{C_k - 1} \sum_{n=m+1}^{C_k} cos(e_m^k, e_n^k). \quad (4)$$

● **Inter-task ER (Inter-ER).** This term promotes geometric independence between embeddings across different tasks:

$$\mathcal{L}_{Inter} = \frac{1}{\sum_{k=1}^{K-1} \sum_{l=k+1}^K C_k C_l} \sum_{k=1}^{K-1} \sum_{l=k+1}^K \sum_{m=1}^{C_k} \sum_{n=1}^{C_l} |cos(e_m^k, e_n^l)|. \quad (5)$$

For Intra-ER, as discussed in (Yu et al., 2024a), greater inter-class distance of spurious features enhances attack performance. This distance can be expressed as $\|e_m^k - e_n^k\|_2^2 = \|e_m^k\|_2^2 + \|e_n^k\|_2^2 - 2\|e_m^k\|_2 \|e_n^k\|_2 \cdot cos(e_m^k, e_n^k)$. To ensure the model effectively learns the introduced features, it is important to enlarge the inter-class distance for each task. However, simply increasing the norm of $e_m^k$ doesn't suffice, as the decoder $D(\cdot; \phi_D)$ can rescale the weights. Thus, minimizing cosine similarity between embeddings is better.

For Inter-ER, geometric independence offers several benefits: (1) Minimized Redundancy: It helps minimize redundancy in the features, and the decoder can exploit the unique information carried by each feature. (2) Reduced Coupling: It helps reduce coupling, and the decoder can focus on each feature independently, leading to more accurate

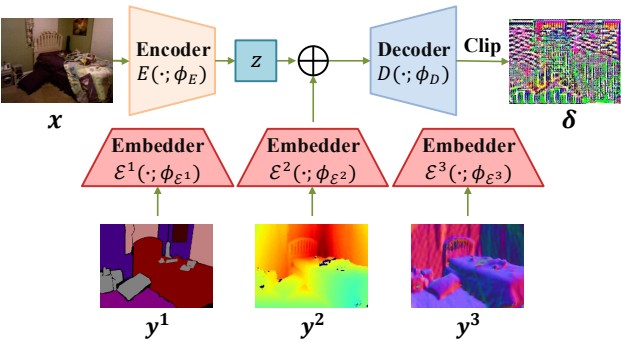

*Figure 4.* MTL-UE applied to dense prediction tasks, *e.g.,* NYUv2.

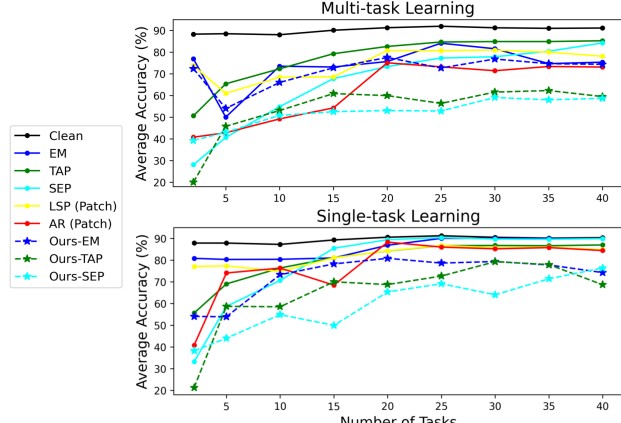

*Figure 5.* Performance Vs. the number of tasks on the CelebA.

perturbations. (3) Improved Interpretability: Geometric independence facilitates understanding of the role each feature plays in generating spurious features.

**Optimization of MTL-UE.** The generator optimization in Alg. 1 can be applied to surrogate-dependent UE methods. After optimization, we transform clean $\mathcal{T}$ to unlearnable $\mathcal{P}$.

**Advantages.** MTL-UE offers key advantages over baselines. First, by incorporating task label priors through embeddings, we reduce the perturbation search space from $\|\boldsymbol{\delta}\|_\infty \le \frac{8}{255}$ to the decoder's output space, leading to lower intra-class variance. Second, since the generator is trained across the entire dataset, it captures global features more effectively, supporting the effective learning of spurious features. Additionally, both Intra-ER and Inter-ER are introduced to further improve attack performance.

### 4.4. Application to Dense Prediction Tasks

In this section, we demonstrate the effective application of MTL-UE to multi-task datasets for dense prediction tasks, using the NYUv2 dataset (Nathan Silberman & Fergus, 2012) as an example. As shown in Fig. 4, instead of using class-wise embeddings for each task, we apply the embedding module $\mathcal{E}^k(\cdot; \phi_{\mathcal{E}^k})$ to map the corresponding dense label $\boldsymbol{y}^k$ to spurious features. Since MTL-UE on such dataset requires manipulating dense prediction results, where redundancy in spurious features is minimal, we do not employ embedding regularizations here.

## 5. Experiments

### 5.1. Experimental Setup

**Datasets.** We choose 4 popular multi-task vision datasets: CelebA (Liu et al., 2015), ChestX-ray14 (Wang et al., 2017), UTKFace (Zhang et al., 2017), and NYUv2 (Nathan Silberman & Fergus, 2012). CelebA has 202,599 face images with 40 binary attribute classifications. We use 162,770 images for training and 19,962 for testing, resizing all to

70×70. ChestX-ray14 has 112,000 chest X-ray images with 14 binary classes for thoracic diseases. The official splits are used, with images resized to 256×256. UTKFace has 23,705 images annotated for age, gender, and race. We follow Karkkainen & Joo (2021) to treat age as a nine-class task, gender as binary, and race as five-class, splitting 80% for training and 20% for testing, with images resized to 140×140. NYUv2 is an indoor scene dataset, with 795 training and 654 testing images for tasks like 13-class semantic segmentation, depth estimation, and surface normal prediction. We follow Liu et al. (2019) to resize images to 288 × 384.

**Models.** We use ResNet-18 (He et al., 2016) as the shared encoder for both surrogate and target models. To evaluate transferability, we also include various backbones like ResNet-50, VGG16 (Simonyan & Zisserman, 2015), DenseNet-121 (Huang et al., 2017), and ViT-B (Dosovitskiy et al., 2021) for the target models. For MTL models, we employ the HPS (Caruana, 1993) architecture, which includes a shared feature extractor and task-specific heads.

**Task-weighting for MTL.** We adopt LS with a uniform weight. To assess transferability, we include Random Loss Weighting (RLW) (Lin et al., 2022), Uncertainty Weighting (UW) (Kendall et al., 2018), Aligned-MTL (Senushkin et al., 2023), and FairGrad (Ban & Ji, 2024).

**Unlearnable examples.** We include baseline methods EM (Huang et al., 2021), TAP (Fowl et al., 2021), and SEP (Chen et al., 2023) with an $\ell_\infty = \frac{8}{255}$ bound, and LSP (Yu et al., 2022a) and AR (Sandoval-Segura et al., 2022) with an $\ell_2 = 1$ bound, following default settings. Adaptations for MTL are detailed in Sec. 4.2. MTL-UE is compatible with EM, TAP, and SEP, resulting in MTL-UE-EM, MTL-UE-TAP, and MTL-UE-SEP, all using the same $\ell_\infty = \frac{8}{255}$ bounds. More details of the baselines are in Sec. A.1.

**Model and MTL-UE training.** Details for training the

*Table 2.* Results on classification datasets: We report average accuracy (%) for CelebA and UTKFace, and AUC-ROC for ChestX-ray14. For UTKFace, accuracy for age, race, and gender is also shown. All models use ResNet-18, and MTL models use LS for task weighting.

| Dataset → | CelebA (Liu et al., 2015) | | ChestX-ray14 (Wang et al., 2017) | | UTKFace (Zhang et al., 2017) | | | | | | | |
| Model → | MTL | STL | MTL | STL | MTL | | | | STL | | | |
| Tasks → | Avg.↓ | Avg.↓ | Avg.↓ | Avg.↓ | Age↓ | Race↓ | Gender↓ | Avg.↓ | Age↓ | Race↓ | Gender↓ | Avg.↓ |
|---|---|---|---|---|---|---|---|---|---|---|---|---|
| Clean | 91.11 | 90.35 | 0.7577 | 0.6493 | 60.32 | 84.07 | 92.51 | 78.97 | 60.46 | 84.45 | 91.86 | 78.92 |
| LSP (Patch) (Yu et al., 2022a) | 78.12 | 84.80 | 0.6467 | 0.6543 | 18.40 | 17.51 | 49.83 | 28.58 | 19.62 | 15.04 | 62.62 | 32.43 |
| AR (Patch) (Sandoval-Segura et al., 2022) | 73.12 | 84.41 | 0.5306 | 0.6118 | 9.70 | 19.66 | 52.87 | 27.41 | 19.16 | 13.92 | 47.17 | 26.75 |
| LSP (Average) (Yu et al., 2022a) | 91.07 | 90.35 | 0.7218 | 0.6452 | 12.59 | 19.72 | 52.17 | 28.16 | 23.21 | 44.20 | 70.34 | 45.91 |
| AR (Average) (Sandoval-Segura et al., 2022) | 91.14 | 90.37 | 0.7259 | 0.6400 | 13.92 | 41.03 | 55.42 | 36.79 | 14.26 | 42.89 | 52.85 | 36.67 |
| EM (Huang et al., 2021) | 75.66 | 89.91 | 0.4976 | 0.5548 | 19.24 | 17.43 | 58.57 | 31.74 | 25.74 | 37.36 | 89.54 | 50.88 |
| TAP (Fowl et al., 2021) | 85.24 | 87.00 | 0.5478 | 0.6005 | 25.86 | 39.28 | 52.74 | 39.29 | 32.15 | 59.35 | 86.58 | 59.36 |
| SEP (Chen et al., 2023) | 84.25 | 89.91 | 0.5462 | 0.5926 | 25.42 | 44.03 | 52.64 | 40.70 | 33.73 | 60.59 | 88.04 | 60.78 |
| MTL-UE-EM | 74.38 | 74.26 | **0.4813** | **0.5302** | 10.00 | **12.78** | 54.73 | 25.84 | 9.66 | **12.47** | 56.81 | 26.32 |
| MTL-UE-TAP | 59.51 | **68.65** | 0.5341 | 0.6091 | 9.49 | 18.23 | **31.81** | **19.84** | 15.97 | 19.03 | **41.75** | **25.59** |
| MTL-UE-SEP | **58.73** | 76.39 | 0.4929 | 0.6068 | **7.28** | 16.20 | 40.08 | 21.19 | **7.26** | 21.84 | 55.61 | 28.24 |

*Table 3.* Intra-class std of the features for competing UE methods.

| UEs → | EM | TAP | SEP | LSP (P) | AR (P) | MTL-UE-EM | MTL-UE-TAP | MTL-UE-SEP |
|---|---|---|---|---|---|---|---|---|
| Avg. | 2.191 | 3.656 | 6.719 | 1.828 | 1.733 | 1.715 | 2.130 | 2.387 |
| Max. | 82.13 | 103.12 | 96.24 | 34.84 | 20.59 | 18.04 | 47.93 | 64.57 |

MTL/STL models and MTL-UE are given in Sec. A.2.

**Metrics.** We use accuracy for CelebA and UTKFace, and AUC-ROC for ChestX-ray14 (Lin et al., 2022; Achituve et al., 2024). For NYUv2, we use mean Intersection over Union (mIoU) and pixel accuracy (PAcc) for segmentation, absolute errors (AErr) and relative errors (RErr) for depth estimation, and mean absolute error (Mean) and median absolute error (MED) for normal estimation (Liu et al., 2019).

## 5.2. Experimental Results

We first evaluate MTL-UE on CelebA and ChestX-ray14 for binary classifications. We then demonstrate its effectiveness on UTKFace with more than two categories, and finally, show its generalization to the NYUv2, which includes dense predictions for both classification and regression tasks.

**Results on the CelebA.** As shown in Tab. 2, MTL-UE consistently improves surrogate-dependent UE methods like EM, TAP, and SEP. Notably, for STL models, MTL-UE-TAP achieves around 68% accuracy, much better than the baseline, while for MTL models, MTL-UE-TAP and MTL-UE-SEP achieve accuracies below 60%. While MTL-UE-EM has minimal impact on MTL models, it significantly boosts performance on STL models. Fig.5 shows UE performance as a function of task number, highlighting the effectiveness of MTL-UE across varying task counts. MTL-UE shows consistent results for MTL models when the task number exceeds 10. For STL models, it performs well with around 15 tasks, while performance under more tasks remains an area for future exploration. Tab. 3 demonstrates that MTL-UE significantly reduces the intra-class standard deviation of the features, resulting in improved attack performance.

**Results on the ChestX-ray14.** As shown in Tab. 2, MTL-UE-EM excels, and MTL-UE-TAP and MTL-UE-SEP improve upon TAP and SEP. As TAP and SEP use adversarial examples, the low performance of clean surrogate models lowers the results, explaining why MTL-UE-EM is better.

**Results on the UTKFace.** The results in Tab. 2 show that with 3 tasks, MTL-UE consistently outperforms the baselines on both MTL and STL models, despite the increased class numbers. Notably, on STL models, MTL-UE reduces average accuracies by over 30% compared to the base UEs. Sec. B.1 presents UE performance vs. task numbers.

**Results on the NYUv2.** We show the results on the NYUv2 involving dense classification and regression tasks. From Tab. 4, MTL-UE consistently enhances the performance of all baseline UE across all tasks. Each variant—MTL-UE-EM, MTL-UE-TAP, and MTL-UE-SEP—exhibits different trade-offs across the three tasks: MTL-UE-EM performs best in segmentation, MTL-UE-TAP excels in depth estimation, and MTL-UE-SEP outperforms others in surface normal estimation. As AR and LSP cannot be applied to dense tasks, they are not included for comparison.

**Visual results.** We offer visual results in Fig. 6. We can see that compared to baselines, our perturbations are more structured. For instance, MTL-UE-TAP displays clearer semantic information than TAP, with more distinct contours and sketch lines. The more structured perturbations tend to have lower intra-class variance, leading to the improved performance. Among the baselines, AR adopts $\ell_2$-norm, and appears sparser, though the overall perturbation magnitudes are similar across all methods. More results are in Sec. B.5.

## 5.3. Discussion

**Smaller perturbations.** We evaluate smaller bounds $\ell_\infty = \frac{2}{255}, \frac{4}{255}, \frac{6}{255}$ alongside $\ell_\infty = \frac{8}{255}$. Tab. 6 shows MTL-UE are effective, outperforming baselines even with larger bounds. MTL-UE-EM excels on STL models, while the other two perform better on MTL models under small bounds.

*Table 4.* Quantitative results on the NYUv2 dataset. ResNet-18 is used as the encoder, with LS for task weighting in MTL models.

| Model → | MTL | | | | | | STL | | | | | |
|---|---|---|---|---|---|---|---|---|---|---|---|---|
| Task → | Segmentation | | Depth | | Normal | | Segmentation | | Depth | | Normal | |
| Metric → | mIoU↓ | PAcc↓ | AErr↑ | RErr↑ | Mean↑ | MED↑ | mIoU↓ | PAcc↓ | AErr↑ | RErr↑ | Mean↑ | MED↑ |
| Clean | 53.05 | 75.01 | 0.3920 | 0.1665 | 23.72 | 17.29 | 53.16 | 75.30 | 39.80 | 16.50 | 22.53 | 16.00 |
| EM | 26.45 | 46.22 | 0.6109 | 0.2352 | 30.32 | 24.46 | 27.03 | 44.66 | 0.6280 | 0.2426 | 29.09 | 22.53 |
| TAP | 21.96 | 36.40 | 0.6412 | 0.2561 | 33.79 | 27.87 | 14.51 | 26.32 | 0.6340 | 0.2389 | 30.75 | 23.25 |
| SEP | 11.41 | 23.49 | 0.7113 | 0.2884 | 37.11 | 31.52 | 9.76 | 24.16 | 0.7517 | 0.2774 | 33.01 | 26.14 |
| MTL-UE-EM | **2.37** | **16.04** | 0.9249 | 0.3013 | 38.04 | 33.65 | **1.65** | **15.64** | 0.9109 | 0.2978 | 33.98 | 27.49 |
| MTL-UE-TAP | 16.88 | 30.37 | **0.9459** | **0.3161** | 39.69 | 34.61 | 17.14 | 30.01 | **1.1182** | **0.3607** | **41.56** | **35.08** |
| MTL-UE-SEP | 17.76 | 33.13 | 0.8260 | 0.2850 | **41.78** | **36.47** | 15.35 | 33.44 | 0.9570 | 0.3188 | 40.24 | 33.08 |

*Figure 6.* Visual results: Odd rows show perturbations (independently normalized to [0,1]), and even rows show poisoned images.

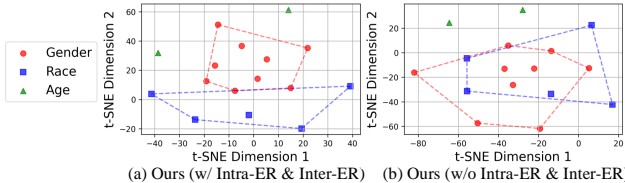

*Figure 7.* T-SNE results of the learned class-wise embeddings.

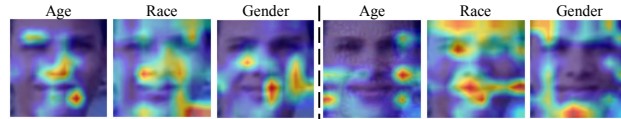

*Figure 8.* GradCAM of models trained on clean/unlearnable data.

**Transferability.** We assess UE transferability to MTL and STL models with varying encoder backbones and MTL weighting strategies. Results on CelebA are shown in Tab. 5. Our methods, particularly MTL-UE-TAP and MTL-UE-SEP, generalize well across CNNs, while MTL-UE-EM performs better on ViT-B. This may be because SEP and TAP rely on adversarial examples, limiting transferability from CNNs to ViT, as ResNet-18 was used as the surrogate backbone. Our methods also generalize well to MTL models with different weighting strategies, with MTL-UE-SEP excelling in several cases. Additional results on NYUv2, along with MTL architecture transfer, are provided in Sec. B.4.

**Ablation study (design & $\lambda_1$, $\lambda_2$).** We do an ablation study of the MTL-UE design, with results in Tab. 7. Compared to ①, ② introduces an auto-encoder for perturbation gener-

ation, but shows little impact, even reducing performance on MTL models. ③, which relies solely on learnable class-wise feature embeddings, shows a significant performance improvement over ①, emphasizing the role of embedding priors in reducing intra-class variance. Additionally, building on ③, ours incorporates latents $z$ from the input $x$, and further enhances performance, likely due to improved optimization by leveraging additional information from $x$. Finally, ④ excludes intra-task and ⑤ excludes inter-task embedding regularization, both showing that embedding regularization enhances performance. We also provide an ablation study of hyperparameters $\lambda_1$ and $\lambda_2$ in Sec. B.2, showing that MTL-UE is not sensitive to them.

**Partial task protection.** We explore the scenario, where only selected tasks are unlearnable, using MTL-UE-SEP. After optimizing the generator, we generate UE by using the mean of $\{e_i^k\}_{i=1}^{C_k}$ for learnable tasks instead of $e_{y^k}^k$. Results in Tab. 8 show that for STL models, unprotected tasks match clean data accuracy, while protected tasks perform like those

*Table 5.* Results of Transferability: performance on CelebA when transferring to MTL and STL models with different backbones or to MTL models with various weighting strategies. Note that RN denotes the ResNet, and DN denotes the DenseNet.

| | Transfer to MTL and STL models with various **backbone** | | | | | | | | | | | | Transfer to MTL models with various **task-weighting** | | | | | |
| Model → | MTL (LS as the task-weighting) | | | | | | STL | | | | | | MTL(ResNet-18 as the backbone | | | | | |
| Backbone/Weighting → | RN-18 | RN-50 | VGG-16 | DN-121 | ViT-B | Avg. | RN-18 | RN-50 | VGG-16 | DN-121 | ViT-B | Avg. | LS | UW | RLW | Align | FairGrad | Avg. |
|---|---|---|---|---|---|---|---|---|---|---|---|---|---|---|---|---|---|---|
| Clean | 91.11 | 91.20 | 91.33 | 91.37 | 89.39 | 90.88 | 90.35 | 90.13 | 90.40 | 89.61 | 87.25 | 89.55 | 91.11 | 90.82 | 91.00 | 91.08 | 90.75 | 90.95 |
| LSP (Patch) | 78.12 | 76.37 | 77.96 | 77.07 | 81.96 | 78.70 | 84.80 | 84.26 | 82.08 | 84.70 | 86.96 | 84.56 | 78.12 | 77.48 | 72.95 | 77.43 | 73.80 | 75.56 |
| AR (Patch) | 73.12 | 73.07 | 72.23 | 75.49 | 85.73 | 75.53 | 84.41 | 85.40 | 75.77 | 78.19 | 87.03 | 82.16 | 73.12 | 71.87 | 74.59 | 72.59 | 67.76 | 71.99 |
| LSP (Average) | 91.07 | 91.15 | 91.31 | 91.32 | 89.16 | 90.60 | 90.35 | 90.08 | 90.56 | 89.77 | 87.01 | 89.55 | 91.07 | 90.73 | 90.98 | 90.89 | 90.62 | 90.86 |
| AR (Average) | 91.14 | 91.20 | 64.78 | 67.09 | 89.35 | 80.31 | 90.37 | 90.12 | 89.47 | 88.15 | 87.16 | 89.05 | 91.14 | 90.80 | 91.01 | 90.93 | 90.69 | 90.91 |
| EM | 75.66 | 74.26 | 72.29 | 76.20 | 83.76 | 76.83 | 89.91 | 89.50 | 75.87 | 86.93 | 86.93 | 85.03 | 75.66 | 75.30 | 75.32 | 75.34 | 74.54 | 75.23 |
| TAP | 85.24 | 85.44 | 87.34 | 87.65 | 88.58 | 86.45 | 87.00 | 86.74 | 85.38 | 84.57 | 86.83 | 86.10 | 85.24 | 85.27 | 85.52 | 85.23 | 84.94 | 85.24 |
| SEP | 84.25 | 81.27 | 83.10 | 82.81 | 89.31 | 84.55 | 89.91 | 89.68 | 84.93 | 86.52 | 86.33 | 87.47 | 84.25 | 89.54 | 87.33 | 87.28 | 87.78 | 87.24 |
| MTL-UE-EM | 74.38 | 72.81 | 69.81 | 70.31 | **71.12** | 71.69 | 74.26 | 74.52 | 78.78 | 76.68 | 79.03 | 76.25 | 74.38 | 70.84 | 72.71 | 71.78 | 73.25 | 72.59 |
| MTL-UE-TAP | 59.51 | 64.76 | **60.50** | 60.36 | 76.69 | **64.76** | 68.65 | 70.73 | 68.27 | 75.14 | **83.22** | **73.20** | 59.51 | 58.66 | 62.95 | 63.60 | 56.61 | 60.27 |
| MTL-UE-SEP | **58.73** | **60.06** | 64.61 | 63.71 | 80.14 | 65.85 | 76.39 | 79.87 | 70.17 | 75.25 | 84.78 | 77.69 | **58.73** | 53.54 | 58.24 | 60.54 | **53.60** | **56.53** |

*Table 6.* Results of MTL-UE with smaller bounds on CelebA.

| Bound → | $\ell_\infty = 8/255$ | | $\ell_\infty = 6/255$ | | $\ell_\infty = 4/255$ | | $\ell_\infty = 2/255$ | |
| Model → | MTL | STL | MTL | STL | MTL | STL | MTL | STL |
|---|---|---|---|---|---|---|---|---|
| MTL-UE-EM | 74.38 | 74.26 | 74.38 | 75.11 | 75.56 | **76.25** | 78.81 | **79.24** |
| MTL-UE-TAP | 59.51 | **68.65** | 63.69 | **72.58** | 63.91 | 80.01 | **69.61** | 85.19 |
| MTL-UE-SEP | **58.73** | 76.39 | **60.98** | 80.45 | **61.82** | 83.58 | 81.35 | 86.82 |

*Table 8.* Results of partial task protection on on the UTKFace.

| Model → | | MTL | | | STL | | |
| Protected task ↓ | | Age | Race | Gender | Age | Race | Gender |
|---|---|---|---|---|---|---|---|
| None | | 60.32 | 84.07 | 92.51 | 60.46 | 84.45 | 91.86 |
| Age, Race | | 12.68 | 20.53 | 88.52 | 10.25 | 19.64 | 90.95 |
| Age, Gender | | 10.72 | 78.82 | 43.90 | 10.70 | 82.68 | 46.41 |
| Race, Gender | | 52.64 | 35.49 | 56.79 | 57.11 | 16.52 | 52.97 |
| All | | 7.28 | 16.20 | 40.08 | 7.26 | 21.84 | 55.61 |

*Table 7.* Ablation study of the MTL-UE design on CelebA.

| Model → | | MTL | | | STL | | |
| Base UE → | | EM | TAP | SEP | EM | TAP | SEP |
|---|---|---|---|---|---|---|---|
| ①: the baseline UE | | 75.66 | 85.24 | 84.25 | 89.91 | 87.00 | 89.91 |
| ②: w/o the feature embeddings | | 90.75 | 88.08 | 86.73 | 89.95 | 88.81 | 87.78 |
| ③: w/o the $E(\cdot; \phi_E)$ and $z$ | | 74.47 | 62.37 | 68.63 | 78.82 | 71.06 | 78.45 |
| ④: w/o the Intra-ER | | 77.87 | 63.07 | 63.41 | 79.82 | 69.52 | 79.25 |
| ⑤: w/o the Inter-ER | | 75.81 | 64.95 | 63.69 | 78.42 | 76.05 | **75.40** |
| MTL-UE | | **74.38** | **59.51** | **58.73** | **74.26** | **68.65** | 76.39 |

*Table 9.* Results of partial data protection on the NYUv2.

| $r$ → | 0% | 20% | | 40% | | 60% | | 80% | | 100% |
| | $\mathcal{T}_1$ | $\mathcal{P}_{0.2}+\mathcal{T}_{0.8}$ | $\mathcal{T}_{0.8}$ | $\mathcal{P}_{0.4}+\mathcal{T}_{0.6}$ | $\mathcal{T}_{0.6}$ | $\mathcal{P}_{0.6}+\mathcal{T}_{0.4}$ | $\mathcal{T}_{0.4}$ | $\mathcal{P}_{0.8}+\mathcal{T}_{0.2}$ | $\mathcal{T}_{0.2}$ | $\mathcal{P}_1$ |
|---|---|---|---|---|---|---|---|---|---|---|
| PAcc | 75.0 | 74.0 | 74.5 | 72.3 | 71.1 | 69.9 | 67.4 | 63.5 | 60.8 | 33.1 |
| RErr | 0.167 | 0.167 | 0.169 | 0.173 | 0.180 | 0.183 | 0.199 | 0.194 | 0.237 | 0.285 |
| Mean | 23.7 | 24.2 | 24.3 | 24.7 | 25.6 | 25.3 | 27.4 | 29.0 | 32.8 | 41.8 |

**Feature visualization on UTKFace with MTL-UE-TAP.** Fig.7 shows t-SNE results of the learned class-wise embeddings for each task, where Inter-ER improves task separation and Intra-ER disperses embeddings within tasks. Fig. 8 shows GradCAM results for MTL models on their training samples, revealing that MTL-UE introduces spurious features, shifting focus to irrelevant areas like facial contours instead of key regions like eyes and noses.

trained to protect all tasks. In contrast, MTL models show degraded performance for all tasks, likely due to shared encoder learning both benign and spurious features.

**Partial data protection.** Following Huang et al. (2021), we convert a portion $r$ of clean $\mathcal{T}$ into unlearnable $\mathcal{P}_r$, leaving the rest as $\mathcal{T}_{1-r}$. We experiment on the NYUv2 using MTL-UE-SEP. Table 9 shows results for models trained on $\mathcal{P}_r$ mixed with $\mathcal{T}_{1-r}$ and only on $\mathcal{T}_{1-r}$. The effectiveness drops quickly when the data is not fully unlearnable, a limitation also noted in Huang et al. (2021). Models trained on the mixture or only partial clean data show similar results, indicating that $\mathcal{P}_r$ is ineffective and unlearnable during training.

**Computational cost & Parameter count.** Sec. A.3 shows MTL-UE's efficiency is close to EM and much lower than TAP and SEP, and LSP and AR are faster due to predefined patterns. Sec. A.4 shows MTL-UE needs fewer parameters than EM, TAP, and SEP, but slightly more than AR and LSP.

**Resistance to defenses.** Sec. B.3 shows MTL-UE is much more robust to SOTA defenses than baseline UE, likely due to its clearer semantic patterns, distinct contours, and sketch lines, which better withstand ISS-induced corruptions.

# 6. Conclusion

This paper presents MTL-UE, the first framework for generating UE on multi-task datasets for both MTL and STL models, featuring a plug-and-play design that seamlessly integrates with existing surrogate-dependent methods. Instead of optimizing perturbations for each sample, we utilize an encoder-decoder network with additional sets of class-wise embeddings. By incorporating task label priors through embeddings, MTL-UE reduces the intra-class variance of spurious features for each task. Additionally, the intra-task and inter-task embedding regularization improve the inter-class separation of spurious features and minimize redundancy, further enhancing performance. MTL-UE is also versatile, supporting dense prediction tasks in MTL. Extensive experiments demonstrate the effectiveness of MTL-UE.

## Acknowledgements

This work was carried out at the Rapid-Rich Object Search (ROSE) Lab, Nanyang Technological University (NTU), Singapore. This research is supported by the National Research Foundation, Singapore and Infocomm Media Development Authority under its Trust Tech Funding Initiative, the Basic and Frontier Research Project of PCL, the Major Key Project of PCL, and Guangdong Basic and Applied Basic Research Foundation under Grant 2024A1515010454. Any opinions, findings and conclusions or recommendations expressed in this material are those of the author(s) and do not reflect the views of National Research Foundation, Singapore and Infocomm Media Development Authority.

## Impact Statement

In summary, our paper introduces MTL-UE, a novel framework for generating unlearnable examples (UEs) that protects multi-task learning (MTL) data and tasks from unauthorized exploitation. As MTL models become increasingly important for handling a wide range of tasks simultaneously, safeguarding such models is critical to ensure privacy and prevent the misuse of sensitive data. Our approach addresses the growing need to protect personal or proprietary data from unauthorized use while reducing the risks of data theft in critical sectors like healthcare, finance, and security. Furthermore, it fosters ethical AI development by encouraging responsible data use, promoting trust between data providers and model developers, and contributing to broader conversations on data protection and privacy regulations.

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

# A. More experimental details

## A.1. Details of the baseline UE methods

For experiments on classification datasets, we follow specific setups for both surrogate-dependent and surrogate-free methods:

- **Surrogate-dependent Methods:**
  - For EM (Huang et al., 2021), TAP (Fowl et al., 2021), and SEP (Chen et al., 2023), we use MTL models with Linear Scalarization (LS) as the surrogate model.
  - For TAP and SEP, following their default settings, we employ PGD (Madry et al., 2018) with 250 steps and a step size of $\frac{0.064}{255}$ to generate adversarial perturbations via targeted attacks. The target class for the $k$-th task is set as $(y^k + C_k//2)\%C_k$, where $y^k$ is the original label and $C_k$ represents the total number of classes for the $k$-th task.
  - For SEP, we use an ensemble of 15 checkpoints to enhance attack effectiveness.
  - For EM, perturbations are optimized using PGD with 20 steps and a step size of $\frac{0.8}{255}$. Surrogate models are trained for 10 iterations after each loop of perturbation optimization. The stopping criterion, as per the default setting, requires the overall training set accuracy to reach 99%.

- **Surrogate-free Methods:**
  - For LSP (Yu et al., 2022a) and AR (Sandoval-Segura et al., 2022), we evaluate two fusion strategies:
    * *Averaging-based Fusion:* For each $k$-th task, $C_k$ class-wise perturbations with the same shape as the input are generated. These are then fused using the averaging-based approach.
    * *Patch-based Fusion:* We divide the input into $N \times N$ patches, where $N = 2^{\lceil \log_2(\lceil K^{0.5} \rceil) \rceil}$. For an input shape of $3 \times H \times W$, the patch size becomes $3 \times H//N \times W//N$. Perturbations for the $K$ tasks are placed in corresponding patches (the first $K$ patches) based on their labels $y^k$. These perturbations are generated separately for $C_k$ classes in each patch for each $k$-th task, and then the whole perturbations are interpolated from shape $3 \times N * (H//N) \times N * (W//N)$ to match the input shape $3 \times H \times W$.
  - It is worth noting that for both LSP and AR, the perturbation bound is set to $\ell_2 = 1$ regarding the inputs with a shape of $32 \times 32$, as per their default settings. However, for larger input sizes, such as $H \times W$, the bound is adjusted to $\ell_2 = 1 \times \frac{H}{32} \times \frac{W}{32}$ to ensure comparability with other methods.

For experiments on the NYUv2 dataset, we include only surrogate-dependent methods:

- For EM (Huang et al., 2021), TAP (Fowl et al., 2021), and SEP (Chen et al., 2023), we use MTL models with Linear Scalarization (LS) as the surrogate model.

- For TAP and SEP, following their default settings, PGD (Madry et al., 2018) is employed to generate adversarial perturbations via targeted attacks. For the semantic segmentation task, the target class is defined as $(y^k + 13//2)\%13$, given there are 13 semantic classes in total. For the depth estimation and surface normal estimation tasks, where defining a target is challenging, untargeted attacks are used, aiming to maximize the loss between the attacked prediction and the ground truth labels.

- For SEP, an ensemble of 10 checkpoints is used to enhance attack effectiveness.

- For EM, perturbations are optimized using PGD, while surrogate models are trained for 10 iterations after each loop of perturbation optimization. The stopping criterion is reached when the loop of perturbation optimization completes 10 iterations, resulting in a total of 200 iterations for each perturbation optimization (as each PGD step runs for 20 iterations).

## A.2. Details of the models and the training

**Design of MTL-UE generator.** Assume the input shape is set to $3 \times H \times W$ or $1 \times H \times W$ for Grayscale images such as the ChestX-ray-14 dataset, and there are $K$ tasks. For the design of MTL-UE's generator, we use 9 convolutional layers and $\times 0.5$ downsampling as $E(\cdot; \phi_E)$, resulting in $z$ with shape $128 \times \frac{H}{2} \times \frac{W}{2}$. For the set $\left\{ \{e_i^k\}_{i=1}^{C_k} \right\}_{k=1}^{K}$, each $e_{y^k}^k$ is designed to match the length and width of $z$, giving it the shape $16 \times \frac{H}{2} \times \frac{W}{2}$. For $D(\cdot; \phi_D)$, we use 4 ConvTranspose2d layers and one $\times 2$ upsampling. The input to $D(\cdot; \phi_D)$ is the concatenation of $z$ and $\{e_{y^k}^k\}_{k=1}^{K}$, resulting in a shape of $(128 + 16 \times K) \times \frac{H}{2} \times \frac{W}{2}$, and the output is the same shape as the input to $E(\cdot; \phi_E)$.

Specifically, for the NYUv2 dataset, we use the embedder $\mathcal{E}^k(\cdot; \phi_{\mathcal{E}^k})$ instead of $e_{y^k}^k$. The structure of the embedder is identical to that of the corresponding $E(\cdot; \phi_E)$, with the input shape for segmentation labels and depth being $1 \times H \times W$, and the input for normal labels being $3 \times H \times W$.

**MTL and STL model training.** For the CelebA, ChestX-ray14, and UTKFace datasets, we train the models for 60 epochs using the Adam optimizer, starting with a learning rate of 1e-3. We apply the MultiStepLR scheduler, with milestones at epochs 36 and 48, and a gamma of 0.1 to adjust the learning rate. The batch size is set to 512 for CelebA and UTKFace, and 128 for ChestX-ray14. Data augmentation techniques include RandomCrop and RandomHorizontalFlip to improve the model's generalization. For the NYUv2 dataset, we train the models for 200 epochs, starting with a learning rate of 1e-4, and use a StepLR scheduler with a step size of 100 and a gamma of 0.1 to reduce the learning rate during training. We set the batch size to 8 for this dataset due to the larger image dimensions and the computational load involved.

**MTL-UE generator training.** The batch sized, training epochs, learning rates, optimizers, and learning rate schedulers for the generator are configured identically to those used for training the MTL models on the respective datasets. For the hyperparameters in Alg. 1, $\epsilon$ is set to match the baseline methods, with the default value $\epsilon = \frac{8}{255}$ for all datasets. The weight $\lambda_1$ is set to 20, and $\lambda_2$ is set to 100 across all datasets. For MTL-UE-EM, where the "Train-surrogate" is set to *True*, we use 10 iterations, consistent with the baseline EM methods. For MTL-UE-TAP and MTL-UE-SEP, where "Train-surrogate" is set to *False*, we pretrain the MTL models on the respective datasets using the default training settings as described earlier. For the remaining configurations, we follow the same setup as the corresponding baseline methods: EM, TAP, and SEP.

## A.3. Computational Complexity

We also analyze the computational complexity of MTL-UE and the baseline methods. For both EM and MTL-UE-EM, the computational complexity is nearly identical since they share the same stopping criterion. For TAP and SEP, each perturbation requires 250 optimization steps on each surrogate model. In contrast, for MTL-UE-TAP and MTL-UE-SEP, perturbations are optimized once per epoch, with a total of 60 epochs for classification datasets and 200 epochs for the NYUv2 dataset, which means 60 optimization steps on each sample for classification dataset, and 200 optimization steps on each sample for NYUv2 dataset. This makes our methods more efficient compared to the baseline TAP and SEP. For AR and LSP, the perturbations are predefined and hand-crafted, making these methods the most computationally efficient.

## A.4. Parameter Quantity

We provide the parameter counts for all competing methods:

- For EM, TAP, and SEP, the parameters correspond to the perturbations. For example, on the CelebA dataset, the perturbation parameters have a shape of $162770 \times 3 \times 70 \times 70 = 2,392,719,000$.

- For our method, the parameters are derived from the MTL-UE generator, which consists of 3,086,147 parameters for the CelebA dataset.

- For AR and LSP, the parameter count for the averaging-based fusion method is $2 \times 40 \times 3 \times 70 \times 70 = 1,176,000$, while for the patch-based fusion method, it is $2 \times 40 \times 3 \times 8 \times 8 = 15,360$.

Overall, our method is efficient, requiring only slightly more parameters than the patch-based AR and LSP methods.

*Table 10.* Quantitative results on the UTKFace with different number of tasks: We report the average accuracy (%) across the selected tasks. Both MTL and STL models adopt ResNet-18 as the encoder. The MTL models employ LS as the task-weighting.

| Selected tasks → | Age | Age, Race | | Age, Race, Gender | |
|---|---|---|---|---|---|
| Model → | MTL/STL | MTL | STL | MTL | STL |
| Metric → | Avg.↓ | Avg.↓ | Avg.↓ | Avg.↓ | Avg.↓ |
| Clean | 57.51 | 70.79 | 70.45 | 78.97 | 78.92 |
| LSP (Yu et al., 2022a) (Patch) | 10.65 | 13.11 | 18.27 | 28.58 | 32.43 |
| AR (Sandoval-Segura et al., 2022) (Patch) | 9.47 | 14.60 | 14.61 | 27.41 | 26.75 |
| LSP (Yu et al., 2022a) (Average) | 10.65 | 18.23 | 30.73 | 28.16 | 45.91 |
| AR (Sandoval-Segura et al., 2022) (Average) | 9.47 | 12.34 | 14.07 | 36.79 | 36.67 |
| EM (Huang et al., 2021) | 10.89 | 19.16 | 19.26 | 31.74 | 50.88 |
| TAP (Fowl et al., 2021) | 25.82 | 31.57 | 49.10 | 39.29 | 59.36 |
| SEP (Chen et al., 2023) | 22.74 | 32.63 | 48.86 | 40.70 | 60.78 |
| MTL-UE-EM | 18.82 | 10.60 | **8.82** | 25.84 | 26.32 |
| MTL-UE-TAP | 9.18 | **9.74** | 11.92 | **19.84** | **25.59** |
| MTL-UE-SEP | **5.49** | 11.78 | 12.61 | 21.19 | 28.24 |

*Table 11.* Ablation study of the hyperparameters of MTL-UE on CelebA.

| Model → | MTL | | | STL | | |
|---|---|---|---|---|---|---|
| Base UE → | EM | TAP | SEP | EM | TAP | SEP |
| ⑥: $\lambda_1 = 2, \lambda_2 = 100$ | 75.04 | 60.53 | 59.39 | 73.88 | 69.67 | 75.43 |
| ⑦: $\lambda_1 = 200, \lambda_2 = 100$ | 74.27 | 59.60 | 59.06 | 75.13 | 67.49 | 76.02 |
| ⑧: $\lambda_1 = 20, \lambda_2 = 10$ | 73.85 | 58.86 | 58.81 | 73.29 | 67.84 | 77.08 |
| ⑨: $\lambda_1 = 20, \lambda_2 = 1000$ | 74.23 | 60.02 | 58.09 | 74.95 | 68.98 | 75.82 |
| MTL-UE ($\lambda_1 = 20, \lambda_2 = 100$) | 74.38 | 59.51 | 58.73 | 74.26 | 68.65 | 76.39 |

*Table 12.* Results under state-of-the-art defenses. We select the UTKFace dataset and train the MTL models.

| Defense → | None | | | | ISS-JPEG (Liu et al., 2023) | | | | ISS-Grayscale (Liu et al., 2023) | | | | ISS-BDR (Liu et al., 2023) | | | |
|---|---|---|---|---|---|---|---|---|---|---|---|---|---|---|---|---|
| Tasks → | Age↓ | Race↓ | Gender↓ | Avg.↓ | Age↓ | Race↓ | Gender↓ | Avg.↓ | Age↓ | Race↓ | Gender↓ | Avg.↓ | Age↓ | Race↓ | Gender↓ | Avg.↓ |
| Clean | 60.32 | 84.07 | 92.51 | 78.97 | 59.26 | 83.92 | 92.24 | 78.47 | 59.45 | 83.19 | 91.77 | 78.14 | 58.69 | 82.45 | 92.32 | 77.82 |
| LSP (Patch) | 18.40 | 17.51 | 49.83 | 28.58 | 44.51 | 68.48 | 87.78 | 66.93 | 15.80 | 22.26 | 53.00 | 30.35 | 34.56 | 41.84 | 78.92 | 51.77 |
| AR (Patch) | 9.70 | 19.66 | 52.87 | 27.41 | 47.07 | 82.55 | 90.99 | 73.54 | 27.41 | 41.35 | 52.72 | 40.49 | 14.64 | 26.24 | 52.78 | 31.22 |
| LSP (Average) | 12.59 | 19.72 | 52.17 | 28.16 | 49.81 | 80.76 | 90.86 | 73.81 | 16.98 | 54.81 | 56.50 | 42.76 | 32.41 | 65.70 | 78.23 | 58.78 |
| AR (Average) | 13.92 | 41.03 | 55.42 | 36.79 | 57.87 | 83.40 | 92.07 | 77.78 | 6.46 | 19.28 | 52.83 | 26.19 | 24.54 | 48.06 | 81.16 | 51.25 |
| EM | 19.24 | 17.43 | 58.57 | 31.74 | 58.38 | 82.49 | 91.86 | 77.57 | 19.81 | 24.81 | 58.84 | 34.49 | 19.81 | 24.81 | 58.84 | 34.49 |
| TAP | 25.86 | 39.28 | 52.74 | 39.29 | 46.69 | 78.27 | 89.18 | 71.38 | 24.20 | 56.54 | 69.37 | 50.04 | 24.20 | 56.54 | 69.37 | 50.04 |
| SEP | 25.42 | 44.03 | 52.64 | 40.70 | 47.09 | 78.69 | 89.51 | 71.77 | 23.06 | 49.03 | 57.97 | 43.35 | 23.06 | 49.03 | 57.97 | 43.35 |
| MTL-UE-EM | 10.00 | **12.78** | 54.73 | 25.84 | **6.75** | **14.05** | 57.78 | 26.20 | 7.72 | **16.31** | 57.62 | 27.22 | 12.93 | 22.51 | 63.35 | 32.93 |
| MTL-UE-TAP | 9.49 | 18.23 | **31.81** | 19.84 | 15.82 | 26.27 | **26.39** | 22.83 | 10.68 | 16.73 | **40.27** | 10.68 | 15.02 | 24.73 | 26.39 | 22.05 |
| MTL-UE-SEP | **7.28** | 16.20 | 40.08 | 21.19 | 15.23 | 39.83 | 43.27 | 32.78 | **5.86** | 28.78 | 45.04 | 26.56 | 18.02 | 34.14 | 42.93 | 31.69 |

# B. Additional Results

## B.1. More quantitative results

**Performance of UE Vs. the number of tasks.** Additional results on the UTKFace dataset with varying numbers of tasks are presented in Tab. 10. Even with a limited number of tasks, MTL-UE consistently outperforms the baseline approaches.

## B.2. More ablation study

We do additional ablation studies on MTL-UE, focusing on the hyperparameters $\lambda_1$ and $\lambda_2$ in Alg. 1. Specifically, experiments are conducted on the CelebA dataset. The results in Tab. 11 indicate that MTL-UE is relatively insensitive to these hyperparameters. This may be because $\mathcal{L}_{\text{Intra}}$ and $\mathcal{L}_{\text{Inter}}$ are easier to optimize within the combined loss framework in Alg. 1. As long as $\lambda_1$ and $\lambda_2$ are not set too small, these hyperparameters have minimal impact on the optimization process of MTL-UE.

## B.3. Results to defenses

We evaluate MTL-UE and competing methods against existing UE defenses applicable to MTL models. Experiments are conducted on the UTKFace dataset using MTL models. We select the SOTA defense, Image Shortcut Squeezing (ISS) (Liu et al., 2023), which includes three preprocessing-based techniques: JPEG compression (quality set to 10), grayscale conversion, and bit-depth reduction (BDR, depth set to 2), follwoing their default settings. Results in Tab. 12 show that MTL-UE remains significantly more robust and consistent than baseline UE methods. Notably, under JPEG compression, all baseline methods fail, whereas MTL-UE maintains strong performance. This robustness is likely due to MTL-UE's clearer semantic patterns, distinct contours, and sketch lines, which better withstand ISS-induced corruptions.

*Table 13.* **Backbone transfer:** Performance on the NYUv2 (Nathan Silberman & Fergus, 2012) when transfering to MTL and STL models with various backbone. The MTL models employ HPS (Caruana, 1993) as architecture and linear scalarization as the task-weighting strategy.

| Backbone → | ResNet-101 | | | | | | ResNet-152 | | | | | | WideResNet-50 | | | | | |
|---|---|---|---|---|---|---|---|---|---|---|---|---|---|---|---|---|---|---|
| Task → | Segmentation | | Depth | | Normal | | Segmentation | | Depth | | Normal | | Segmentation | | Depth | | Normal | |
| Metric → | mIoU↓ | PAcc↓ | AErr↑ | RErr↑ | Mean↑ | MED↑ | mIoU↓ | PAcc↓ | AErr↑ | RErr↑ | Mean↑ | MED↑ | mIoU↓ | PAcc↓ | AErr↑ | RErr↑ | Mean↑ | MED↑ |
| Clean | 54.38 | 76.00 | 0.3802 | 0.1582 | 23.16 | 16.79 | 54.88 | 76.29 | 0.3690 | 0.1502 | 22.71 | 16.33 | 52.42 | 74.55 | 0.4044 | 0.1746 | 24.14 | 17.80 |
| EM | 31.78 | 50.72 | 0.5634 | 0.2076 | 28.25 | 22.41 | 32.51 | 50.03 | 0.5395 | 0.1990 | 28.09 | 22.09 | 29.08 | 45.71 | 0.6352 | 0.2246 | 30.71 | 25.84 |
| TAP | 37.73 | 56.71 | 0.4905 | 0.1908 | 27.59 | 21.25 | 32.96 | 50.58 | 0.5176 | 0.1922 | 28.22 | 21.97 | 37.98 | 57.13 | 0.5051 | 0.2016 | 29.41 | 23.26 |
| SEP | 27.85 | 40.81 | 0.5457 | 0.2151 | 30.61 | 24.97 | 29.78 | 43.91 | 0.5789 | 0.2061 | 30.11 | 24.18 | 30.73 | 45.99 | 0.5425 | 0.2175 | 29.81 | 23.61 |
| MTL-UE-EM | 2.38 | 16.03 | 0.8887 | 0.2903 | 38.23 | 33.47 | 1.75 | 15.82 | 0.9638 | 0.3062 | 34.91 | 29.19 | 2.34 | 16.16 | 0.8448 | 0.2812 | 36.71 | 31.71 |
| MTL-UE-TAP | 23.25 | 37.00 | 0.9739 | 0.3203 | 37.56 | 31.66 | 24.10 | 35.71 | 0.8993 | 0.2942 | 36.66 | 30.25 | 23.59 | 36.66 | 0.8403 | 0.2853 | 38.72 | 32.87 |
| MTL-UE-SEP | 18.55 | 29.78 | 0.9043 | 0.3046 | 42.07 | 36.44 | 19.31 | 25.59 | 0.8499 | 0.2835 | 37.64 | 31.72 | 17.38 | 29.48 | 0.8095 | 0.2836 | 41.15 | 35.84 |

*Table 14.* **Weighting strategies transfer:** Performance on the NYUv2 (Nathan Silberman & Fergus, 2012) when transfering to MTL and STL models with various task-weighting strategies. The MTL models employ HPS (Caruana, 1993) as architecture and ResNet-18 (He et al., 2016) as encoder's backbone.

| Weighting → | UW | | | | | | RLW | | | | | | Align | | | | | | FairGrad | | | | | |
|---|---|---|---|---|---|---|---|---|---|---|---|---|---|---|---|---|---|---|---|---|---|---|---|---|
| Task → | Segmentation | | Depth | | Normal | | Segmentation | | Depth | | Normal | | Segmentation | | Depth | | Normal | | Segmentation | | Depth | | Normal | |
| Metric → | mIoU↓ | PAcc↓ | AErr↑ | RErr↑ | Mean↑ | MED↑ | mIoU↓ | PAcc↓ | AErr↑ | RErr↑ | Mean↑ | MED↑ | mIoU↓ | PAcc↓ | AErr↑ | RErr↑ | Mean↑ | MED↑ | mIoU↓ | PAcc↓ | AErr↑ | RErr↑ | Mean↑ | MED↑ |
| Clean | 52.99 | 74.77 | 0.3941 | 0.1696 | 23.76 | 17.33 | 52.70 | 74.90 | 0.3906 | 0.1622 | 23.43 | 16.92 | 52.50 | 74.24 | 0.3966 | 0.1707 | 22.96 | 16.37 | 52.21 | 74.32 | 0.3927 | 0.1694 | 22.85 | 16.34 |
| EM | 25.70 | 43.96 | 0.6124 | 0.2312 | 30.23 | 24.30 | 28.64 | 50.19 | 0.6407 | 0.2391 | 30.57 | 24.94 | 26.01 | 44.12 | 0.5997 | 0.2281 | 29.53 | 23.15 | 28.03 | 48.94 | 0.6367 | 0.2314 | 30.02 | 23.72 |
| TAP | 21.02 | 35.33 | 0.6273 | 0.2532 | 33.76 | 27.86 | 19.79 | 34.79 | 0.6449 | 0.2485 | 34.22 | 27.78 | 18.88 | 31.75 | 0.6261 | 0.2441 | 33.06 | 26.68 | 21.12 | 36.06 | 0.6314 | 0.2382 | 32.95 | 26.29 |
| SEP | 12.20 | 24.40 | 0.6996 | 0.2823 | 36.77 | 31.19 | 11.40 | 23.67 | 0.7048 | 0.2899 | 35.88 | 30.20 | 12.36 | 24.72 | 0.6882 | 0.2813 | 35.98 | 29.73 | 13.47 | 25.78 | 0.6812 | 0.2766 | 35.16 | 29.59 |
| MTL-UE-EM | 2.33 | 15.96 | 0.9515 | 0.3110 | 38.33 | 34.14 | 2.05 | 15.79 | 1.0405 | 0.3344 | 37.03 | 32.16 | 1.78 | 15.74 | 1.0002 | 0.3265 | 37.34 | 31.58 | 2.07 | 15.82 | 0.9987 | 0.3240 | 35.84 | 29.91 |
| MTL-UE-TAP | 17.08 | 28.98 | 0.9620 | 0.3177 | 40.03 | 34.48 | 16.85 | 29.16 | 1.0631 | 0.3489 | 40.96 | 35.68 | 18.31 | 29.53 | 1.0100 | 0.3310 | 38.17 | 31.78 | 17.93 | 30.83 | 0.9513 | 0.3169 | 38.78 | 32.89 |
| MTL-UE-SEP | 13.96 | 21.71 | 0.8600 | 0.2936 | 41.19 | 35.85 | 15.64 | 28.85 | 0.9047 | 0.3057 | 42.49 | 37.15 | 16.50 | 26.67 | 0.8981 | 0.3024 | 41.85 | 35.67 | 16.58 | 26.36 | 0.9020 | 0.3017 | 40.59 | 34.87 |

*Table 15.* **Architecture transfer:** Performance on the NYUv2 (Nathan Silberman & Fergus, 2012) when transfering to MTL and STL models with various architecture. The MTL models employ ResNet-18 (He et al., 2016) as encoder's backbone and linear scalarization as the task-weighting strategy.

| Backbone → | Cross-stitch | | | | | | MTAN | | | | | | LTB | | | | | |
|---|---|---|---|---|---|---|---|---|---|---|---|---|---|---|---|---|---|---|
| Task → | Segmentation | | Depth | | Normal | | Segmentation | | Depth | | Normal | | Segmentation | | Depth | | Normal | |
| Metric → | mIoU↓ | PAcc↓ | AErr↑ | RErr↑ | Mean↑ | MED↑ | mIoU↓ | PAcc↓ | AErr↑ | RErr↑ | Mean↑ | MED↑ | mIoU↓ | PAcc↓ | AErr↑ | RErr↑ | Mean↑ | MED↑ |
| Clean | 52.48 | 74.74 | 0.3874 | 0.1621 | 23.34 | 16.52 | 53.99 | 75.44 | 0.3806 | 0.1619 | 23.34 | 16.67 | 51.53 | 74.30 | 0.3819 | 0.1585 | 23.44 | 16.54 |
| EM | 24.03 | 42.11 | 0.6582 | 0.2377 | 30.29 | 23.96 | 27.04 | 45.61 | 0.6241 | 0.2334 | 30.32 | 24.03 | 28.55 | 50.00 | 0.6357 | 0.2334 | 30.04 | 23.65 |
| TAP | 21.05 | 38.89 | 0.6622 | 0.2455 | 33.99 | 27.18 | 19.72 | 30.56 | 0.6468 | 0.2484 | 34.06 | 27.94 | 18.24 | 32.63 | 0.6527 | 0.2472 | 33.57 | 26.86 |
| SEP | 9.80 | 21.18 | 0.7687 | 0.2814 | 36.03 | 29.83 | 10.53 | 21.45 | 0.7049 | 0.2854 | 36.06 | 30.25 | 10.53 | 21.98 | 0.7679 | 0.2793 | 36.46 | 30.47 |
| MTL-UE-EM | 1.78 | 15.77 | 1.0116 | 0.3249 | 35.18 | 29.46 | 1.54 | 15.66 | 0.9962 | 0.3211 | 36.00 | 30.29 | 1.52 | 15.67 | 1.0654 | 0.3408 | 36.16 | 30.67 |
| MTL-UE-TAP | 17.34 | 30.07 | 1.0652 | 0.3457 | 39.06 | 32.83 | 14.85 | 25.33 | 1.1076 | 0.3587 | 42.24 | 36.80 | 20.44 | 36.43 | 1.0231 | 0.3309 | 38.96 | 32.33 |
| MTL-UE-SEP | 15.78 | 30.89 | 0.9215 | 0.3099 | 42.77 | 36.85 | 14.94 | 26.49 | 0.8888 | 0.3015 | 43.73 | 38.62 | 17.87 | 30.27 | 0.9095 | 0.3036 | 40.71 | 34.88 |

## B.4. More transferability results for NYUv2 dataset

Transferability results for the NYUv2 dataset are shown in Tab. 13, Tab. 14, and Tab. 15. In addition to backbone and weighting strategy transfer on the CelebA dataset, we also evaluate architecture transfer for NYUv2. For backbone transfer, we use ResNet-101, ResNet-152, and WideResNet-50 as backbones for victim models. For weighting strategy transfer, we adopt the same strategies as in the CelebA experiments. For architecture transfer, Cross-stitch (Misra et al., 2016), MTAN (Liu et al., 2019), and LTB (Guo et al., 2020) are included for evaluation. As observed, MTL-UE consistently protects against unauthorized model training on the NYUv2 dataset across various model backbones, weighting strategies, and MTL architectures.

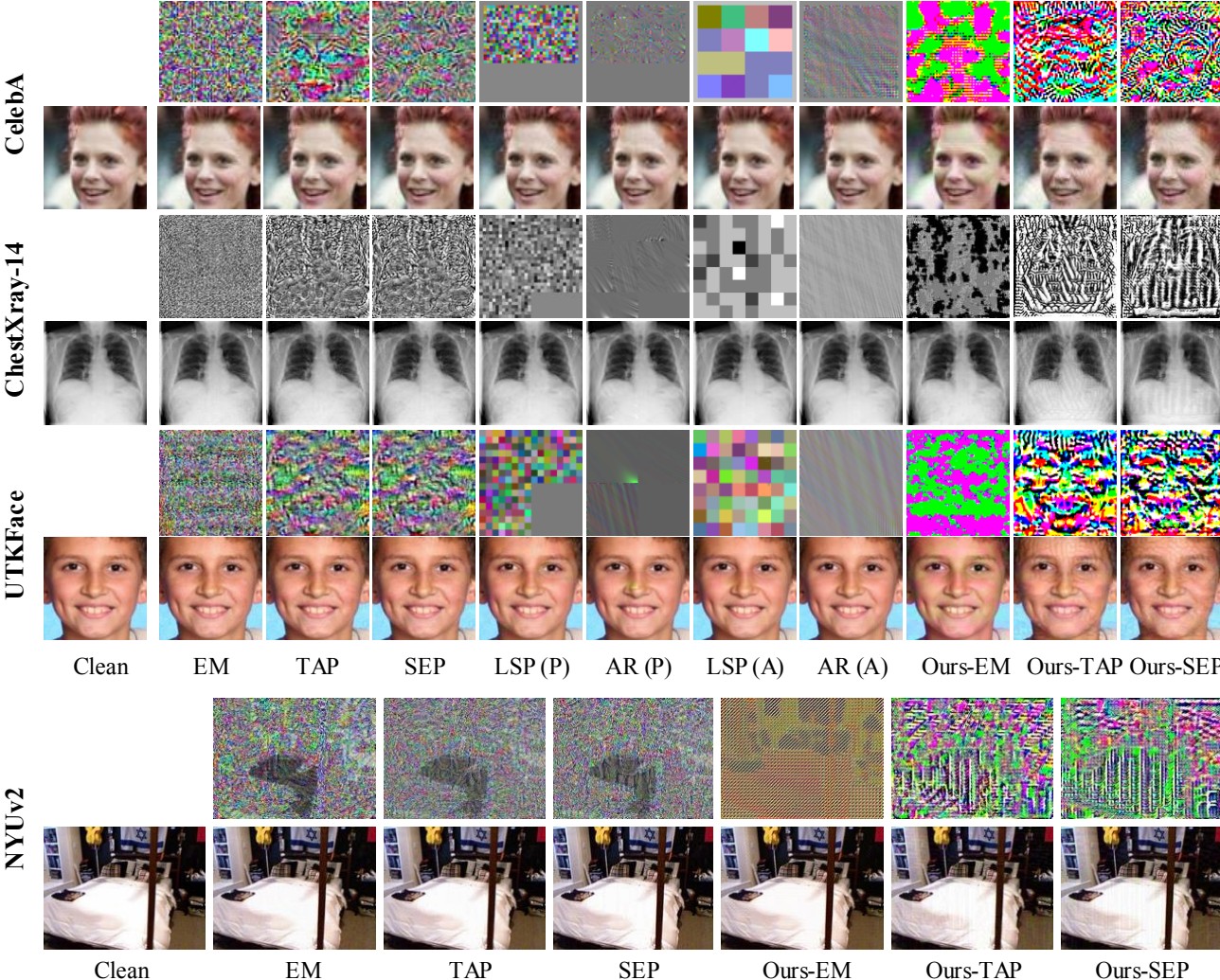

*Figure 9.* Visual results: the odd/even rows show the perturbations/images, respectively. Perturbations are in dependently normalized to [0,1] for clarity.

## B.5. More visual results

We provide additional visual examples of perturbations and poisoned images from the CelebA (Liu et al., 2015), ChestX-ray14 (Wang et al., 2017), UTKFace (Zhang et al., 2017), and NYUv2 (Nathan Silberman & Fergus, 2012) datasets, as shown in Fig. 9. We observe across all datasets that our methods offer more interpretable cues, guiding the model to learn the added perturbations rather than the benign features of the clean data. In particular, MTL-UE-TAP and MTL-UE-SEP display distinct outlines or contour lines.

