# OpenReview forum: "MTL-UE: Learning to Learn Nothing for Multi-Task Learning"
_ICML.cc/2025/Conference — ICML 2025 poster_

### Official Review · Reviewer_Aqty · 2025-03-01

**Overall Recommendation:** 4

**Summary:**

This paper introduces MTL-UE, the first unified framework for creating unlearnable examples tailored for multi-task data and models. By leveraging a generator-based structure with label priors and class-wise embeddings, MTL-UE enhances attack robustness through intra-task and inter-task regularization. It supports dense prediction tasks and integrates seamlessly with existing unlearnable methods, demonstrating superior performance across diverse datasets, models, and task-weighting strategies.

## update after rebuttal
Thanks for the author's reply. All my concerns have been addressed, so I still recommend that this paper be accepted.

**Claims And Evidence:**

The paper presents MTL-UE as an effective framework for generating unlearnable examples in multi-task learning (MTL), and its claims are well-supported by extensive experiments. The results across multiple datasets, architectures, and task-weighting strategies demonstrate that MTL-UE consistently outperforms existing unlearnable example methods and that its embedding-based perturbation strategy effectively reduces intra-class variance, strengthening its core contributions.
While the evidence is strong, a few areas could be further clarified. For example, the paper highlights MTL-UE’s plug-and-play compatibility with existing UE methods, but additional discussion on potential implementation challenges would be helpful.
Overall, I find it sufficiently clear; even though I encountered some concerns while reading, I later discovered that the authors addressed and discussed these issues in the subsequent text.

**Essential References Not Discussed:**

I believe that the significant references have already been encompassed by the authors within the discussion.

**Experimental Designs Or Analyses:**

The experimental setup is thorough, evaluating MTL-UE across datasets, architectures, and task configurations. The inclusion of both MTL and STL models ensures a comprehensive assessment, and the choice of task-weighting strategies is relevant for understanding shared representations.
The ablation study on intra-task and inter-task embedding regularization clearly clarifies their contributions. However, further analysis of how perturbations affect different task types within the same dataset (e.g., classification vs. regression in NYUv2) would add insight. This paper also includes results for smaller perturbations, which enhance imperceptibility while maintaining the unlearnability.

**Methods And Evaluation Criteria:**

The proposed method and evaluation criteria are well-suited for protecting MTL data from unauthorized use.
The evaluation is rigorous, covering various task complexities with datasets for binary classification (CelebA, ChestX-ray14) and multi-class/dense prediction tasks (UTKFace, NYUv2).
The inclusion of multiple MTL task-weighting strategies strengthens the study, as different weightings impact task interactions.
The paper also assesses transferability across architectures, though results show slightly weaker performance with ViTs, warranting further exploration.

**Other Comments Or Suggestions:**

Including more examples of partial task protection, where only one task needs to be protected, would be beneficial. It would also be insightful to explore how MTL performs when the number of tasks to be protected is fewer than the unprotected tasks. Overall, I think this manuscript is good and fully deserves to be accepted by the ICML community.
Tables 11 and 12 are missing references in Sections B.2 and B.3, which should be addressed.

**Other Strengths And Weaknesses:**

Strengths:
This paper provides extensive analysis and explores additional scenarios. Beyond the main results, it includes feature visualizations, with Fig. 7 illustrating how the embedding regularizations enhance feature separation. Additionally, the paper evaluates its robustness against SOTA defenses.
The concept of partial data and partial task protection is particularly intriguing, as it enhances the method's practicality and applicability in real-world scenarios.
This work covers classification and dense prediction tasks across multiple datasets.
Weakness:
Additional discussion on scenarios involving fine-tuning (using a pretrained feature encoder) and advanced augmentations would be valuable.
A line-by-line description of Algorithm 1 would provide readers with a better understanding.

**Questions For Authors:**

See weakness and comments as above.

**Relation To Broader Scientific Literature:**

This paper primarily focuses on benchmarking datasets such as CIFAR-10. While the methods presented could be applied to other scientific problems, their applicability would require further exploration and discussion.

**Theoretical Claims:**

The paper leans on intuitive justifications rather than formal theoretical claims. The key argument is that embedding regularization (Intra-ER and Inter-ER) enhances perturbation effectiveness by controlling feature space alignment. This is reasonable and supported mainly by empirical evidence rather than formal proofs.
A theoretical perspective on why MTL-UE improves attack transferability across tasks and architectures could strengthen the work. The generator-based approach is argued to enable stronger feature-level control, but a formal analysis would reinforce this claim. Additionally, bounding the perturbation space with embeddings is a novel choice, and further discussion on its impact on optimization dynamics could add depth.

---

> ### Author Rebuttal · Authors · 2025-03-28
>
> **Q1**. Scenarios involving fine-tuning (using a pretrained feature encoder) and advanced augmentations.
>
> **A1**. Thanks for the suggestions. We conducted experiments on the proposed scenarios. The table presents results of training MTL models on UTKFace with ImageNet-pretrained-encoder fine-tuning or UEraser [1] as advanced augmentations. Fine-tuning showed no impact on any method, while strong augmentations affected all, with our attacks still outperforming baselines.
>
> |            | None      | Finetuning | UEraser   |
> | ---------- | --------- | ---------- | --------- |
> | Clean      | 78.97     | 79.10      | 79.84     |
> | LSP (P)    | 28.58     | 27.37      | 69.31     |
> | AR (P)     | 27.41     | 23.64      | 54.23     |
> | LSP (A)    | 28.16     | 33.99      | 79.21     |
> | AR (A)     | 36.79     | 36.98      | 74.96     |
> | EM         | 31.74     | 34.03      | 73.19     |
> | TAP        | 39.29     | 40.21      | 61.80     |
> | SEP        | 40.70     | 40.87      | 61.29     |
> | MTL-UE-EM  | 25.84     | 30.37      | 52.50     |
> | MTL-UE-TAP | **19.84** | 21.08      | **32.36** |
> | MTL-UE-SEP | 21.19     | **20.18**  | 38.47     |
>
>
> **Q2**. Line-by-line description of Algorithm 1.
>
> **A2**. Thanks for your suggestions! We’ll add a detailed description in the updated version.
>
> **Q3**. Examples of partial task protection, where only one task is protected.
>
> **A3**. Following Table 8, we show single-task protection results. We can see that for STL models, unprotected tasks match clean data accuracy, while protected tasks perform like those trained to protect all tasks. Results on MTL models degraded performance, likely due to shared encoder learning both benign and spurious features.
>
> | Model $\rightarrow$                               | MTL                 | STL                 |
> | ------------------------------------------------ | ------------------- | ------------------- |
> | Protected task $\downarrow$; Metric $\rightarrow$ | Age, Race, Gender   | Age, Race, Gender   |
> | None                                             | 60.32, 84.07, 92.51 | 60.46, 84.45, 91.86 |
> | Age                                              | 20.84, 80.51, 89.60 | 13.90, 82.78, 90.30 |
> | Race                                             | 55.70, 31.35, 91.35 | 57.47, 17.45, 90.57 |
> | Gender                                           | 56.31, 81.14, 60.59 | 57.28, 82.51, 50.38 |
> | All                                              | 7.28, 16.20, 40.08  | 7.26, 21.84, 55.61  |
>
> **Q4**. Tables 11 and 12 are missing references in Sections B.2 and B.3, which should be addressed.
>
> **A4**. Thanks for your suggestions. We will fix it in the updated version.
>
>
>
> [1] Learning the unlearnable: Adversarial augmentations suppress unlearnable example attacks. ICCVW 2023

---

> > ### Comment · Reviewer_Aqty · 2025-04-02
> >
> > Thanks for the author's reply. All my concerns have been addressed, so I still recommend that this paper be accepted.

---

> > > ### Author Response · Authors · 2025-04-03
> > >
> > > We sincerely appreciate the reviewer’s positive feedback and recognition of the contributions in our paper.

---

### Official Review · Reviewer_AfLP · 2025-03-09

**Overall Recommendation:** 5

**Summary:**

This paper introduces MTL-UE, the first framework for generating unlearnable examples (UEs) tailored for multi-task learning (MTL) models. While existing UE methods focus on single-task learning (STL) to prevent unauthorized training on personal data, modern AI increasingly relies on generalist MTL models. This work addresses this gap by introducing a generator-based approach with class-wise feature embeddings and embedding regularization, improving attack effectiveness and robustness. It supports dense prediction tasks, integrates seamlessly with existing surrogate-dependent UE methods, and enables partial task and data protection. Extensive experiments demonstrate its effectiveness over baseline UE methods across multiple backbones and task-weighting strategies.

**Claims And Evidence:**

Overall, the claims in the paper are well-supported by extensive empirical evidence, including evaluations across multiple datasets, model architectures, and task-weighting strategies.

The authors make several key claims, such as (1) MTL-UE improves attack effectiveness for MTL models, (2) it reduces intra-class variance, (3) it is robust across different datasets and architectures, and (4) it can generalize well to dense prediction tasks.

These claims are backed by thorough experiments on four MTL datasets (CelebA, ChestX-ray14, UTKFace, NYUv2), comparisons against five baseline UE methods, and experiments with five different MTL task-weighting strategies.

**Essential References Not Discussed:**

NA

**Experimental Designs Or Analyses:**

The paper’s experimental design is well-structured for evaluating UEs in MTL. It includes four datasets (CelebA, ChestX-ray14, UTKFace, NYUv2), five task-weighting strategies, and five model architectures, ensuring broad applicability. Meanwhile, the comparison with multiple baselines (LSP, AR, EM, TAP, SEP) provides a fair assessment of MTL-UE.

While the impact of task numbers is analyzed (Figure 5), a deeper exploration of why UEs perform better in MTL than STL would add clarity. Additionally, a direct runtime comparison with baselines would better support claims of computational efficiency.

**Methods And Evaluation Criteria:**

The proposed methods and evaluation criteria align well with generating unlearnable examples (UEs) for multi-task learning (MTL).

The evaluation spans four diverse MTL datasets (CelebA, ChestX-ray14, UTKFace, NYUv2), covering both classification and dense prediction tasks.  Additionally, multiple model backbones from CNNs to Transformers are evaluated to strengthens the claim of transferability.

This work also benchmarks several previous UE methods on multi-tasks learning.

**Other Comments Or Suggestions:**

1.The paper mentions that MTL-UE-TAP and MTL-UE-SEP exhibit lower transferability to models with ViT-B as the backbone because they use ResNet-18 as the surrogate model, resulting in a significant gap between the architectures. How would these methods perform if ViT-B were used as the backbone for the surrogate models?

2.Among the 40 tasks in CelebA, are there specific tasks that are significantly easier or harder to degrade?

**Other Strengths And Weaknesses:**

Strengths:

1.The paper is well-structured and easy to follow, with clear notations and a well-explained methodology.

2.The proposed method is well-justified, with clear relevance to real-world scenarios. This work is interesting and stands out as the first to explore unlearnable examples in the context of multi-task learning.

3.This work initially built a benchmark and identified key weaknesses in existing STL UE methods when applied to MTL.

4.The experimental results are comprehensive, demonstrating effectiveness across multiple datasets and advanced application scenarios.

Weakness:

1.As mentioned in the paper, the performance of MTL-UE on STL could be further enhanced when the number of tasks is very large.  It would be helpful if this paper can give more analysis on the influence of the number of tasks on the UEs performance.

**Questions For Authors:**

Please refer to the comments and suggestions above. Conducting additional experiments is recommended to further validate the findings.

**Relation To Broader Scientific Literature:**

NA

**Theoretical Claims:**

Some theoretical justifications are provided in this work for MTL-UE. The embedding regularization techniques (Intra-ER and Inter-ER) are based on the idea that reducing intra-class variance and increasing inter-class separation enhances perturbation effectiveness. While no formal proofs are given, empirical evidence supports this claim.

MTL-UE’s perturbation generation utilizes class-wise embeddings to constrain the search space, which is theoretically intuitive. While its strong empirical results support the approach, a more in-depth mathematical analysis of its impact on robustness across tasks could further reinforce the findings.

---

> ### Author Rebuttal · Authors · 2025-03-30
>
> **Q1**. How would these methods perform if ViT-B were used as the backbone for the surrogate models?
>
> **A1**. In addition to the results in Table 6, we conducted experiments using ViT-B as the backbone for the surrogate models on the CelebA dataset. The results in the table below show that this choice improves performance when the victim models also use ViT-B but can negatively impact performance when the victim models are CNN-based. This suggests a trade-off in selecting CNNs or ViTs for the surrogate models, depending on the target victim model's backbone.
>
> |Model $\rightarrow$|MTL|STL|
> |-| - | - |
> |Backbone $\rightarrow$|RN-18, RN-50, VGG-16, DN-121, ViT-B, Avg.|RN-18, RN-50, VGG-16, DN-121, ViT-B, Avg.|
> |EM|88.86, 88.75, 84.22, 80.71, 88.67, 86.24|90.11, 89.93, 89.46, 86.42, 86.03, 88.39|
> |TAP|85.55, 86.32, 83.28, 79.38, 86.32, 84.17|88.42, 88.14, 87.97, 87.46, 85.28, 87.45|
> |SEP| 78.67, 81.24, 79.41, 76.55, 79.33, 79.04|87.06, 87.67, 83.36, 83.50, 85.93, 85.50|
> |MTL-UE-EM| 73.68, 73.65, 74.77, 75.85, 74.01, 74.39|78.06, 77.87, 78.72, 77.85, 78.98, 78.29 |
> |MTL-UE-TAP| 68.91, 69.30, 69.40, **66.36**, 73.56, 69.50|76.32, 77.27, 75.13, 70.08, 80.92, 75.94|
> |MTL-UE-SEP| **53.79**, **59.63**, **63.37**, 71.49, **64.01**, **62.45** | **73.12**, **74.45**, **72.71**, **64.45**, **78.66**, **72.67** |
>
> **Q2**. Among the 40 tasks in CelebA, are there specific tasks that are significantly easier or harder to degrade?
>
> **A2**. We compute the accuracy drops for all 40 tasks when training STL models on MTL-UE compared to models trained on clean data. In MTL-UE-EM, tasks 38, 35, and 24 experience the largest drops (-58.74, -58.03, -44.39), while tasks 23, 15, and 39 are least affected (-0.48, -0.70, -1.00). In MTL-UE-TAP, tasks 25, 1, and 40 degrade the most (-80.22, -67.02, -61.90), whereas tasks 36, 38, and 27 show minimal impact (+0.02, 0.00, -0.32). In MTL-UE-SEP, tasks 15, 37, and 2 suffer the highest degradation (-61.94, -49.69, -43.03), while tasks 26, 27, and 14 are least affected (+0.09, +0.02, 0.00). These results suggest that the susceptibility of tasks to degradation varies significantly across different base UE strategies. Tasks experiencing the largest drops may rely on more vulnerable feature representations, making them easier to disrupt. In contrast, tasks with minimal impact likely depend on more robust or less perturbed features.
>
> **Q3**. More analysis on the influence of the number of tasks on the UEs performance.
>
> **A3**. Thank you for the suggestion! We provide a detailed analysis of how the number of tasks affects UE performance. As shown in Figure 5, when training MTL models, increasing the number of tasks to 10 causes only a slight performance drop (minor accuracy increase). From 10 to 40 tasks, MTL-UE remains stable, maintaining ~60% accuracy for both MTL-UE-TAP and MTL-UE-SEP. This stability is likely due to the shared encoder in MTL models (for both the UE generation and victim model training), which facilitates learning spurious features across all tasks, preventing performance degradation.
>
> However, when training STL models with varying task numbers, MTL-UE performance gradually declines as the number of tasks increases from 10 to 20 and 20 to 40. This may result from a mismatch between the UE generation and victim model training, as the former uses MTL surrogate models while the latter adopts STL models. We will include this analysis in the updated version.
>
> **Q4**. A direct runtime comparison with baselines.
>
> **A4**. Thanks for your suggestions. The table below provides a direct runtime comparison for generating perturbations across all competing methods in terms of hours. The generation process was conducted on a single RTX 4090 using the CelebA dataset (with ViT as the backbone of the surrogate MTL model) and the NYUv2 dataset. Notably, LSP and AR, which use predefined patterns, require almost no time for perturbation generation. The results indicate that our MTL-UE has a comparable computational cost to the base UE methods and is sometimes more efficient.
>
> |Method $\downarrow$|CelebA dataset $\downarrow$|NYUv2 dataset $\downarrow$|
> |-|-|-|
> |LSP|Almost 0|-|
> |AR|Almost 0|-|
> |EM|7.35|3.43|
> |TAP|4.81|3.36|
> |SEP|25.01|26.14|
> |MTL-UE-EM|8.55|3.75|
> |MTL-UE-TAP|4.55|2.90|
> |MTL-UE-SEP|22.55|24.87|

---

> > ### Comment · Reviewer_AfLP · 2025-04-03
> >
> > Thanks for the authors' response. After carefully reviewing the comments from the other reviewers and the authors' rebuttal, my concerns have been well addressed. As a result, I have decided to raise my score to 5.

---

> > > ### Author Response · Authors · 2025-04-04
> > >
> > > We sincerely appreciate the reviewer’s positive feedback and recognition of the contributions in our paper.

---

### Official Review · Reviewer_abGT · 2025-03-13

**Overall Recommendation:** 3

**Summary:**

In this paper, we propose an effective method for generating unlearnable samples for multi-task learning (MLT), which uses a generator to generate perturbations instead of the traditional iterative method. In this paper, the effectiveness of the method is analyzed and validated in terms of both accuracy and robustness, and the effectiveness of the method is demonstrated on several datasets.

**Claims And Evidence:**

The authors show the accuracy of existing UE methods in SLT and MLT scenarios in Fig. 2 and find that directly migrating existing UE methods to MLT leads to some degree of accuracy degradation with k increasing. This indicates the poor applicability of the existing methods, thus providing motivation for the proposed method in this paper.

**Essential References Not Discussed:**

No.

**Experimental Designs Or Analyses:**

Experiments are presented on multiple datasets and the results show that the proposed method outperforms existing methods, supporting the authors' conclusions.

**Methods And Evaluation Criteria:**

The proposed method improves the accuracy of UE to some extent. The test datasets include face and medical images, which are selected with practical significance.

**Other Comments Or Suggestions:**

No.

**Other Strengths And Weaknesses:**

Strengths:
The authors identified the problem of low UE accuracy in MLT scenarios and provided an explanation.
The authors proposed an effective UE method and validated it through extensive experiments.

Weaknesses:
1. Insufficient coverage of existing literature. The authors' literature research on data poisoning methods is more limited, covering only up to 2023, while there are still a number of new researches in the field that are worth including. In addition, experimental comparisons mainly focus on UE methods, and fewer comparisons on data-based poisoning fail to clarify its necessity.
2. Challenges of new issues are not significant across datasets. Figure 2 illustrates the task differences between MLT and SLT on the CelebA dataset, but on ChestX-ray14 this difference does not seem to be significant. Especially on the clean data, where 0.75% and 91.1% accuracy are not in the same order of magnitude, it is not quite clear to me why there is such a large difference in task accuracy between the different datasets.
3. Questioning of method validity. The authors provide variants of MLT on three different methods, but the impact of these changes varies dramatically across datasets. For example, the MLT-UE variant on TAP has an even higher average accuracy of the STL model than TAP when testing the ChestX-ray14 dataset. Some of the data suggests that the improvements of MLT-UE are limited and even not always effective in some cases. This contradicts the authors' claim of enhanced attack performance.
4. Insufficient technical novelty. The authors' main improvement is to replace labels with feature embeddings, which is simple and effective but weak in terms of novelty. I expect that the authors can analyze the effectiveness of this improvement from a theoretical perspective, rather than just performing technical implementation and experimental evaluation.
5. The applicability of MLT tasks is unclear. Although the authors include the performance of SLT in their discussion, their motivation for the study seems to be based on the fact that there is some kind of generic challenge between the two. It is not clear to me exactly how this challenge relates to MLT, and I would have appreciated a clearer clarification.
6. The question of the plausibility of attack scenarios. Could the authors please elaborate on the unlearning needs of MLT in real-life scenarios and the scope of application of this paper's methodology, which would help to better understand its practical application value.

**Questions For Authors:**

See weakness

**Relation To Broader Scientific Literature:**

The authors found experimentally that UE performs poorly in multi-task learning and propose improvements accordingly.

**Theoretical Claims:**

The author does not make explicit theoretical claims, so there are no relevant questions.

---

> ### Author Rebuttal · Authors · 2025-03-31
>
> **Q1**. Insufficient coverage of existing literature.
>
> A1. Thank you for the advice! We’ll add recent works on **data poisoning** in the related work. As we focus on UE, we add a comparison between UE and other poisoning attacks (the table below) in the updated paper to broaden the literature coverage and emphasize the need for UE.
> |Aspect|UE|Label Flipping|Backdoor Attacks|Availability Attacks (Partial Poisoning)|Targeted Data Poisoning|
> |-|-|-|-|-|-|
> |**Attack Objective**|Significantly reduce accuracy on clean data, causing near-random guesses|Decrease accuracy for all classes or specific classes|No impact on clean data, but manipulate predictions to a specific class for data with predefined triggers|Reduce accuracy on clean data, but to a lesser degree compared to UE|Misclassify specific target classes or instances|
> |**Nature of Poisoning**|Add subtle, imperceptible perturbations across the entire training dataset and maintain true labels|Modify a portion of dataset by changing labels only|Modify a portion of dataset with triggers (perturbations) and potential label changes|Modify a portion of dataset with unbounded perturbations and potential label changes|Modify a portion of dataset with unbounded perturbations and potential label changes|
>
> **Q2**. Challenges of new issues are not significant across datasets.
>
> A2. Following original settings, we report accuracy (%) for CelebA and UTKFace, while for ChestX-ray14 the metric is AUC-ROC (see Sec. 5.1). The drop in AUC-ROC from 0.7577 to near 0.5 is significant.
>
> **Q3**. Questioning of method validity.
>
> A3. MTL-UE works well with surrogate-dependent UE methods, improving performance across most datasets. The exception is ChestX-ray14 with TAP and SEP, where the clean model performs not well (AUC-ROC=0.75), limiting the success of adversarial attacks and, consequently, the effectiveness of adversarial-attack-based UE methods like TAP and SEP. However, MTL-UE still improves on EM, boosts TAP and SEP in MTL, and matches STL results for TAP and SEP.
>
> **Q4**. Insufficient technical novelty.
>
> A4. MTL-UE innovates in UE for MTL by solving model misalignment with shared task embeddings, boosting STL performance. It optimizes at the distribution level using class-wise embeddings and allows flexible task protection without retraining. Theoretically, we introduce embedding regularizations (Intra-ER & Inter-ER) to maximize intra-task separation and promote inter-task independence, enhancing the learning of spurious features. See A5 for more details on the technical novelty to solve challenges. We’ll refine the theoretical analysis and revise the paper to clarify these.
>
> **Q5**. The applicability of MLT tasks is unclear.
>
> A5. MTL is vital in real-world applications like autonomous vehicles, which handle multiple tasks simultaneously. Research has advanced multi-task datasets and MTL models. While UE has been studied for STL, we are the first to tackle unauthorized training on multi-task data and protect data privacy. To bridge the gap, we adapt UE methods from STL to MTL and highlight key challenges:
> - Model Alignment Issue: GPU constraints limit UE generation to use one MTL model instead of multiple STL models (e.g., 40 for CelebA), causing misalignment when training STL victim models. In MTL-UE, task-wise embeddings align UE within each task, reducing performance degradation when training STL models compared to MTL ones. In CelebA, MTL-UE-EM shows minor MTL gains but major STL improvements.
> - Lack of Distribution-Level Optimization:  Prior UE methods optimize samples independently, ignoring distribution-level effects. In MTL settings like CelebA (40 tasks), this issue worsens. MTL-UE uses class-wise embeddings to poison distributions per task, reducing intra-class variance and reinforcing spurious correlations, enhancing attack effectiveness.
> - Re-optimize for different protected task sets: Prior sample-wise UE methods require re-optimizing perturbations for each combination of tasks to protect. MTL-UE, once optimized on all tasks, allows flexible task selection to protect without re-optimizing (Sec. 5.3).
>
> We'll add these in the paper.
>
>
> **Q6**. The question of the plausibility of attack scenarios.
>
> A6. Unlearning in MTL protects sensitive multi-task data from unauthorized training due to privacy concerns. In medical AI, MTL models use X-rays, MRIs, and patient histories for diagnosis, where unauthorized training risks privacy and ethics violations. Facial recognition models (e.g., CelebA) can enable surveillance and profiling. Social media analysis (e.g., Weibo-20) enables large-scale surveillance. Smart surveillance models (e.g., DukeMTMC) risk privacy infringement. MTL-UE prevents unauthorized use by introducing unlearnable perturbations, degrading MTL and STL model performance, thus enhancing data privacy and intellectual property protection. We'll add these in the paper.
>
> **Q7**. Evaluation of computational complexity.
>
> A7. Please refer to A4 for reviewer AfLP.

---

### Official Review · Reviewer_tZ3V · 2025-03-14

**Overall Recommendation:** 3

**Summary:**

This work studies unlearnable examples (UE) for multi-task learning (MTL). The authors first evaluated baseline UE methods in the MTL scenario, showing that existing UE methods are not effective on MTL when more tasks are involved. Motivated by this observation, MTL-UE is proposed taking both single-task and multi-task learning into account. Comprehensive MTL experiments are provided.

**Claims And Evidence:**

Intra-class variance of optimization-based sample-wise approaches are high, e.g., EM, TAP, and SEP. The explanation makes sense to me. But would it be possible to modify, let’s say  sample-wise EM, to fit the MTL objective by limiting the intra-class variance? EM is designed for STL, so it makes sense it performs badly on MTL. It would be great if the authors could articulate the potential of modifying sample-wise optimization-based approaches to fit MTL. I think the class-wise noise makes the optimization of MTL UE easier, but it's not essential.

**Essential References Not Discussed:**

No.

**Experimental Designs Or Analyses:**

Experiments shown in Figure 2 can be further explained. How to interpret the results on STL with task 40? Does it mean that the UEs are generated on one task and tested on all tasks? Especially, the claim that STL models are more robust to UE than MTL models, can be further clarified. This result is important since patch-based AR outperforms other approaches, which is used as the basis of analysis in the following paper content.

The baseline performance in table 2 can be further clarified. Why is the baseline STL performance also limited? Does it follow the same manner of Figure 2? It would be great if the authors could clarify. I would anticipate that the baseline STL performance should be better than reported.

**Methods And Evaluation Criteria:**

The proposed methods and evaluations make sense. However, the exact evaluation criteria needs to be further clarified. See the comments above regarding STL performance.

**Other Comments Or Suggestions:**

None.

**Other Strengths And Weaknesses:**

None.

**Questions For Authors:**

This work explores the application of unlearnable examples in multi-task learning. The observation is interesting, since STL unlearnable examples do not generalize in the MTL scenario. My concerns listed above can mainly be summarized as the following two points:

1. Can original sample-wise unlearnable examples also be modified to work under the MTL scenario. Except for class-wise noises making the optimization easier, what is the potential obstacle that makes sample-wise unlearnable examples not work?
2. The provided experimental results can be further clarified and interpreted. For example, what does STL mean in Figure 2? Why does baseline perform not well even on STL tasks?

It would be great if the authors could clarify these questions.

**Relation To Broader Scientific Literature:**

None.

**Theoretical Claims:**

Theoretical analysis is not provided in this work.

---

> ### Author Rebuttal · Authors · 2025-03-31
>
> **Q1**. Further clarify the experimental results. What does STL mean in Figure 2, and why do the baseline methods perform poorly, even on STL?
>
> A1. The pipeline for UE has two stages:
>
> - **Stage 1**: UE generation process (Section 4.2).
> - **Stage 2**: UE performance evaluation, where generated UEs train victim models, and these models are tested on clean data.
>
> Stage 1 focuses on generating UE to protect **all tasks simultaneously** using a surrogate MTL model. Stage 2 evaluates in two ways: a) Train one MTL model for all tasks. b) Train individual STL models for each task, and averaging the evaluation results across all tasks.
>
> Figure 2 shows UE performance vs. the task number. For each x-axis point, only the first $K$ tasks are selected for both stages, and UE are generated to protect them. Stage 2 evaluates a) MTL (left) and b) STL (right). Other tables use full task sets with the above stages for a complete UE assessment. All baseline methods designed for STL, perform poorly in Stage 2 for both MTL and STL, as Stage 1 generates UE for MTL models, not STL settings. Figure 2 also shows that when $K$ is small, these baselines perform well but degrade as $K$ increases due to greater misalignment between Stage 1 and Stage 2. We will clarify these in the paper.
>
> ---
>
> **Q2**. Can original sample-wise UE be modified to work under the MTL scenario? Except for class-wise noises making the optimization easier, what is the potential obstacle that makes sample-wise UE not work?
>
> A2. To reduce the high intra-class variance in optimization-based methods, we add additional loss terms. As these methods use a surrogate MTL model during optimization, we define $L_{std}=[L_{std}^1,\ldots,L_{std}^D]$, and $L_{std}^d$ is defined in line 201 of the paper. We use two loss terms, $L_{std}^{mean}$ and $L_{std}^{max}$: the mean and maximum of $L_{std}$. In the perturbation optimization (Eqs. (1) & (2)), we add $\lambda_1 L_{std}^{mean}+\lambda_2 L_{std}^{max}$ to the original loss. If the original optimization is to maximize, we apply a negative sign.
>
> We experiment on CelebA with a batch size of 1024. The table below shows MTL model results with various hyperparameters. As $L_{std}^{mean}\approx5$ and $L_{std}^{max}\approx100$, these choices are reasonable. Despite different settings, the new losses don't outperform the original one. $L_{std}^{max}$ largely degrades performance, and $L_{std}^{mean}$ has a smaller, but still negative, effect.
> |$(\lambda_1,\lambda_2)$|(0,0) in paper|(0.5,0.01)|(0.05,0.001)|(0.05,0)|(0,0.001)|(0.005,0.0001)|(0.005,0)|(0,0.0001)|MTL-UE|
> |-|-|-|-|-|-|-|-|-|-|
> |EM|75.66|90.45|90.57|89.70|90.4|88.05|74.73|87.14|74.38|
> |TAP|85.24|90.84|89.95|88.53|89.79|87.82|85.82|87.73|59.51|
> |SEP|84.25|90.34|90.15|89.25|90.64|89.57|85.37|88.97|58.73|
>
> **Challenges in Adopting These Loss Terms**
>
> - Batch Size Limitation: With 160k images in CelebA and a 1k batch size, variance estimates are unreliable, and increasing batch size is infeasible due to GPU limits.
>
> - Surrogate Model Misalignment: TAP and SEP use a surrogate MTL model trained on clean data, misaligning with the victim model trained on UE. EM partially addresses this by training on UE data, but weak early-stage perturbations lead to suboptimal results.
>
> - Conflict with Original Loss: Minimizing $L_{std}^{mean}$ and $L_{std}^{max}$ conflicts with the original loss. $L_{std}^{max}$ degrades UE effectiveness, and $L_{std}^{mean}$ with small $\lambda_1$ has minimal impact.
>
> - Computational Overhead: These losses increase computation by $\times28$.
>
> **Potential Obstacles for Existing Sample-Wise UE in MTL**
>
> - Model Alignment Issue: Due to GPU limits, UE generation uses one MTL model instead of multiple STL models (e.g., 40 for CelebA), causing misalignment with STL victim models. In MTL-UE, a surrogate MTL model is used, and images with the same label $y^k$ for the $k$-th task share the embedding $e_{y^k}^k$. Thus, MTL-UE suffers less degradation when training STL victim models than MTL models. CelebA results show MTL-UE-EM improves slightly in MTL but shows gains in STL, highlighting its alignment effectiveness.
> - Lack of Distribution-Level Optimization:  Perturbations mislead the victim model by mapping $x_i+\delta_i$ to labels, poisoning the data distribution.
>    - Previous UE methods individually optimize samples, missing distribution-level effects.
>    - EM trains on batched poisoned data, implicitly considering distribution, is better than TAP and SEP.
>    - In MTL settings like CelebA (40 tasks), lack of distribution optimization is problematic.
>    - MTL-UE creates poisoned distributions with class-wise embeddings, reducing intra-class variance and effectively misleading victim models.
> - Re-optimizing for different protected task sets: Prior sample-wise UE methods require re-optimizing perturbations for each combination of tasks to protect. MTL-UE, once optimized on all tasks (Sec. 5.3), allows flexible task selection to protect without re-optimizing.

---

### Decision · Program_Chairs · 2025-05-01

**Decision:**

Accept (poster)

**Comment:**

All reviewers have provided positive scores for this submission, highlighting its strengths in technical contribution, experiments, and workload. Given the unanimous positive feedback and the recognition of its contribution to the area, the AC carefully reviewed the paper and concurred with the reviewers' assessments, therefore supporting the decision to accept this submission.